# Metascientific replication project with the advanced meta-experimental protocol of the transparent psi project procedures for testing the precognitive effect claimed by Bem

Jan Walleczek[1,2]*, Nikolaus von Stillfried[1,2], Stefan Schmidt[3,4], Marc Wittmann[4], Karolina A. Kirmse[4,5], Jorge Moll[2,6], Zoltan Kekecs[7]

1 Phenoscience Laboratories, Berlin, Germany, 2 Paradox Science Institute, Palo Alto, Santa Clara, California, United States of America, 3 Department of Psychosomatic Medicine and Psychotherapy, Medical Faculty, Medical Center, University of Freiburg, Freiburg, Germany, 4 Institute for Frontier Areas of Psychology and Mental Health, Freiburg, Germany, 5 Department of Psychology, Technical University Dresden, Dresden, Germany, 6 Cognitive Neuroscience Unit, D'Or Institute for Research and Education, Rio de Janeiro, Brazil, 7 Institute of Psychology, ELTE, Eotvos Lorand University, Budapest, Hungary

* walleczek@phenoscience.com

## Abstract

This metascientific project studied the replicability of Bem Experiment 1, which had claimed a precognitive effect, i.e., the ability to successfully guess the outcome of future random events (Bem. J Pers Soc Psychol. 2011;100: 407−25). The use of advanced methodologies—based on the advanced meta-experimental protocol (AMP) and transparent psi project (TPP) procedures—reduced the risk of false discoveries as a function of (i) confirmation bias, (ii) non-transparency, and (iii) intrinsic measurement bias. The combined AMP-TPP test strategy performed three replication studies with a total of 26,483 participants resulting in N = 420,472 critical trials. Study 1 failed to replicate the precognitive effect. An exploratory analysis of Study 1 suggested an effect in the opposite direction than was originally predicted (49.48% ± 0.26 SE; N = 37,836). Study 2 confirmed this exploratory result using a high-powered replication design (49.65% ± 0.14 SE; p = 0.013; N = 127,000). Study 3 was unable to replicate the result from Study 2 (50.07% ± 0.11 SE; p = 0.496; N = 217,800). The results of Study 2 represent a rare example in psi research of the successful replication of an exploratory result using a confirmatory protocol. The source of the one-time confirmed anomalous result in Study 2 remains to be identified. This result presents either (i) a psi-derived anomaly that defies known physical laws, or (ii) a method-derived anomaly, e.g., a false-positive statistical finding. Using conventional standards, based on the lack of replicability in Study 3 and absence of an accepted scientific theory, the second scenario appears more plausible. This AMP-TPP metascientific project demonstrated the use of advanced controls for assessing the reliability of the employed scientific process. This project shows how rigor-enhancing

**Data availability statement:** All primary data and analysis codes are publicly available via Open Science Framework (OSF): https://osf.io/8jteq/. There, all data (https://osf.io/68wy2/files) and analysis codes (https://osf.io/k5fnx/files) can be viewed. Preregistered codes and raw datasets in the repository that were originally collected are available through the following links for the individual preregistrations: https://osf.io/s9bqr (Study 1), https://osf.io/3wqk8 (Study 2), https://osf.io/3gtqk (Studies 1 and 2 without participants), and https://osf.io/qkv6p (Study 3). This also includes the following Github repositories: https://github.com/amp-tpp/transparent-psi (AMP-TPP Study), https://github.com/amp2-tpp/transparent-psi (AMP2-TPP Study), and https://github.com/amp4-tpp/transparent-psi (AMP4-TPP Study).

**Funding:** This research was supported in part by the Fetzer Franklin Fund of the John E. Fetzer Memorial Trust (Contract #288), USA (https://www.fetzer-franklin-fund.org), and the Scients Institute (Contract #3), USA (https://scients.org), through collaborative agreements (with SS) at the Institute for Frontier Areas of Psychology and Mental Health (IGGP), Freiburg, Germany (https://www.igpp.de) and supervised by JW of Phenoscience Laboratories, Berlin, Germany (https://www.phenoscience.com). For disclosure, co-author JW was Director of the Fetzer Franklin Fund of the John E. Fetzer Memorial Trust (https://www.fetzer-franklin-fund.org). Co-author JM is Founder and Board Director of the Scients Institute (https://scients.org). Besides these two co-authors (JW, JM), funders had no role in study design, data collection and analysis, decision to publish, or preparation of the manuscript.

**Competing interests:** The authors have declared that no competing interests exist.

test strategies can improve the reliability, not only of psi research, but any type of weak-effects experiments, including in psychology.

---

## 1. Introduction

The continuing search for anomalous cognitive effects in psychology, i.e., parapsychology, has long been criticized (e.g., [1]). Methodological issues, including the use of post-hoc (exploratory) analytic procedures, and the frequent publication of non-replicable results, have prevented the field of parapsychology (i.e., psi research) from progressing beyond the exploratory phase of research—even after investigating psi effects in the laboratory for more than 100 years. These issues also extend to meta-analyses on anomalous-cognition effects involving multiple studies as described in Section 1.2.

During the last decade, investigations in the field of metascience—the science of the scientific process—have revealed that the publication of false-positive or non-replicable findings is far more prevalent in the general scientific literature than was thought previously; therefore, major reform initiatives are underway to increase the reliability and replicability of research findings (e.g., [2–6]). For example, when recent replication studies in parapsychology implemented rigour-enhancing practices, such as confirmatory testing or preregistration, they failed to confirm the replicability of results for well-known psi paradigms. For more details see Section 1.1.

Besides the inability to replicate psi effects and phenomena using credible techniques, another long-time challenge concerns the prohibitive theoretical constraints regarding claimed psi-based interactions between human consciousness and the physical world. For explanation, parapsychologists have long argued that the significant psi effects are the result of anomalous mind-matter interactions and are—therefore—related to research efforts in the study of consciousness (e.g., [7]). The claimed psi effects are called 'anomalous' because if they were proven to exist under controlled laboratory conditions then this would require an explanation beyond known physics. That is, no accepted scientific theory is available for explaining how mind or consciousness could—in fact—overcome the known physical constraints based on the fundamental Laws of Nature. For more information see Section 4.5.

Despite the above-mentioned constraints, interest in psi research has remained strong in some frontier-research communities—often driven by interest in (i) alternative models of reality beyond standard physicalism, and (ii) the advancement of new methodologies for frontier research. Therefore, one major aim of this replication project was to demonstrate how advanced control test strategies can improve the reliability of findings, not only for psi research, but frontier science in general.

### 1.1. The observation of null results with confirmatory protocols in psi research

For a field of research that lacks support by an accepted scientific theory, any progress depends upon the use of credible scientific methodologies for gathering reliable empirical evidence. This includes the use of confirmatory protocols to counter the

experimenter effect called confirmation bias (e.g., [3,8]). Well-known limitations of prior studies in many fields of research, including psi research, concern both (i) the differences between exploratory and confirmatory research designs and (ii) disagreements over what constitutes a valid replication attempt (e.g., [2–17]). When using confirmatory methods and adopting advanced controls, can new insights be gained regarding the true sources, or origins, of the many reports that have claimed (exploratory) evidence for psi effects?

As was alluded to above, when confirmatory methods were used in recent replication studies of popular paradigms in parapsychology, null results were found [18–27]. Obviously, these null results raise the possibility that previously reported anomalous results (as obtained with exploratory approaches) could have—in fact—presented false-positive results. In short, is a given anomalous result the product of a psi-derived anomaly, i.e., a true positive, or method-derived anomaly, i.e., a false positive? At a minimum, given the recent null results with confirmatory replication studies, it should be acknowledged that the true sources of the many positive (exploratory) findings, as reported in the psi literature, remain to be identified on a case-by-case basis. A closely related concern is the low credibility of the claim that meta-analytic studies have already confirmed the overall replicability of psi effects.

## 1.2. The claim of replicable results in psi research based on meta-analytic studies

Meta-analytic investigations of parapsychological studies have often been cited as collective evidence for the overall replicability of parapsychological phenomena in laboratory studies (e.g., [7,28–31]). But how reliable have the control measures been for each individual study in the database of a given meta-analysis? Have these meta-analytic statistical claims considered the insights from the metascience reform movement such as the above-mentioned role of confirmation bias in the results analysis of both individual studies as well as subsequent meta-analyses? For example, the recent meta-analysis by Tressoldi and Storm [31], which claimed an overall positive psi effect, was largely based on exploratory studies. Furthermore, about 40% of the studies included in the database had not been peer-reviewed. This makes it difficult to ascertain whether the findings of the individual studies were not compromised by questionable research practices or allowed for confirmation bias. Finally, meta-analytic statistical analyses can themselves be subject to questionable research practices and confirmation bias. For specific criticisms of meta-analyses in psi research, e.g., see refs. [9,32–34]. Until the insights from the metascience reform movement are adopted, and higher methodological standards are systematically implemented in psi research, the claim of overall replicable psi findings, including based on meta-analytic studies, has low credibility and can easily be challenged.

## 1.3. A metascientific replication project for Bem Experiment 1

The present work investigated the most-widely discussed claim for an experimental psi effect in recent years, namely the precognitive effect claimed with Experiment 1 in the reports by Bem [35] and Bem et al. [36]. The main result of original Bem Experiment 1 was the following: Participants can perform better-than-chance in guessing successfully on which side of a computer screen (left or right) an erotic image will be presented, i.e., significantly above 50% in a binomial statistical test. Bem [35] reported a successful guess rate of 53.1% (p = 0.01). The anomalous result was found only when erotic images were used as a reward, while the guess rate was at chance levels for non-erotic images [35]. Briefly, the reward procedure in Bem Experiment 1 [35] involved the presentation of either an erotic or non-erotic image following the selection by the participant of the correct side (left or right) on the computer screen, i.e., the side where the image will be presented (in the test for precognition). In summary, the present work performed replication studies that included tests for (i) the hypothesis that erotic images lend themselves to precognition and (ii) the hypothesis that non-erotic images do not lend themselves to precognition.

### 1.3.1. The transparent psi project procedures.
To effectively counter the experimenter effect of confirmation bias, this replication project not only adopted confirmatory protocols but also included direct data deposition, i.e., the real-time deposition of data in a version-controlled repository, to make the data collection process fully retraceable [20].

The implementation of these procedures was based on the transparent psi project (TPP) procedures for testing the precognitive effect claimed by Bem which were developed using a consensus-design process involving both sceptics as well as proponents of psi research, including Bem himself [37]. Again, this replication project sought to reproduce the consensus-designed TPP procedures which had previously been employed in a multi-laboratory replication project of Bem Experiment 1 [20]. In this way, a research protocol could be implemented in the present replication project that was widely supported by both skeptics and proponents of psi research. Also directly adopted from the TPP was the overall research methodology, including software programming, database of erotic and non-erotic images, generation of random events, and statistical analysis measures and tools (for details see Section 2.2.1.).

Recognizing that confirmation bias is only one possible confounding influence, other advanced control measures were also implemented to further reduce the risk of false discoveries. For example, there might be sources of systematic error in an experimental study that cannot be revealed, or eliminated, by using preregistered protocols or data transparency. That is, there might operate hidden forms of bias that are intrinsic to the employed scientific process. As will be described next, for the purpose of revealing hidden systematic error sources, a previously developed meta-experimental test strategy was employed as a diagnostic tool [25].

**1.3.2. The advanced meta-experimental protocol.** The advanced meta-experimental protocol (AMP) presents a strictly confirmatory approach for collecting diagnostic information about potential biases that might lead to false conclusions in a study [25]. The AMP uses a meta-experimental test strategy for measuring the absence—or presence— of systematic error sources. In this context, meta-experiments are experiments that test the performance of the used experimental protocol itself. For two examples see Section 1.4., where the AMP-based test strategy of counterfactual meta-experimentation (CFME) is described for capturing diagnostic information about the reliability of the scientific process that is used in this replication project. In general, counterfactual test strategies can empirically investigate the following question: What would have been the result if, for example, a different independent variable, or no independent variable, had been used as part of the employed scientific process? In the present work, as one type of counterfactual strategy, a random event generator (REG) substituted for the participants who are the source of the tested independent variable (see Section 1.4. for details). This counterfactual (REG-based) experiment should report non-significant results unless the scientific process were to produce systematic measurement errors. In short, the CFME-based test strategy seeks to reduce the risk of mistaking false-positive covariations—between outcome measure and the tested intervention— for true-causal ones by confirming empirically the absence of significant systematic error sources.

The AMP-based approach, including the strategy of CFME, allows for the quantitative evaluation of the reliability of a research paradigm at a high level of detail; that is, the AMP can measure up to three different types of systematic error, called class-A, class-B, and class-C error in AMP terminology (see also Section 2.2.2.). A brief overview is provided next: (i) Systematic class-A error presents a bias in the overall scientific process—independent of applying control and experimental conditions. Examples of this type of error are intrinsic imbalances in the measurement procedure, including a potentially biased statistical analysis. (ii) Systematic class-B error presents bias as a function of repeated performances of control and experimental conditions. This type of systematic error might occur, for example, in studies where the same participant performs control or experimental sessions multiple times. For prior work assessing the presence of systematic class-B error in a study see ref. [25]. (iii) Systematic class-C error is bias due to any kind of uncontrolled performance of a control condition. An example of this type of error is a hidden effect of the control condition that competes with the effect of the experimental condition and thereby might produce a false-negative result.

Note that the AMP was originally developed for assessing the possibility of false-positive results in a psi study claiming an anomalous observer-consciousness effect [25,26]. The AMP has helped reveal that prior publications could have easily mistaken false positives for true positives and, therefore, has already contributed to the correction of academic literature [38].

**1.3.3. The combined AMP-TPP test strategy.** By combining the AMP with core elements of the TPP, a highly credible research protocol was implemented in this investigation of a hypothesized anomalous (psi) effect. The combined

AMP-TPP test strategy greatly reduced the potential impact of known short-comings that are often associated with research methodologies in past work. This included methodological limitations as a function of (i) confirmation bias, (ii) non-transparency, i.e., the lack of verifiability of the used research procedures, and (iii) intrinsic measurement bias. Again, this replication project included the scientific investigation of the employed scientific process itself; hence, the term 'metascientific replication project'.

In summary, besides performing a high-powered replication attempt of Bem Experiment 1, a key goal of the AMP-TPP metascientific project was to demonstrate how best scientific practices—as offered by both TPP and AMP procedures—can increase the credibility of experimental research. Furthermore, this work can provide quantitative information regarding the reliability of the consensus-designed TPP procedures as used by Kekecs et al. [20]. Next, the here-employed counterfactual test strategies of the AMP are described.

### 1.4. Two counterfactual test strategies as quality controls with the AMP

To enhance the credibility of new results with Bem Experiment 1, two advanced controls were implemented based on the CFME-based test strategy of the AMP (see also Section 1.3.2.). Again, this enabled the empirical search for hidden measurement biases that could be responsible for false-positive as well as false-negative findings [25]. See Fig 1 for an illustration of the employed research paradigm and explanations of the two CFME-based control strategies as implemented in this replication of Bem Experiment 1.

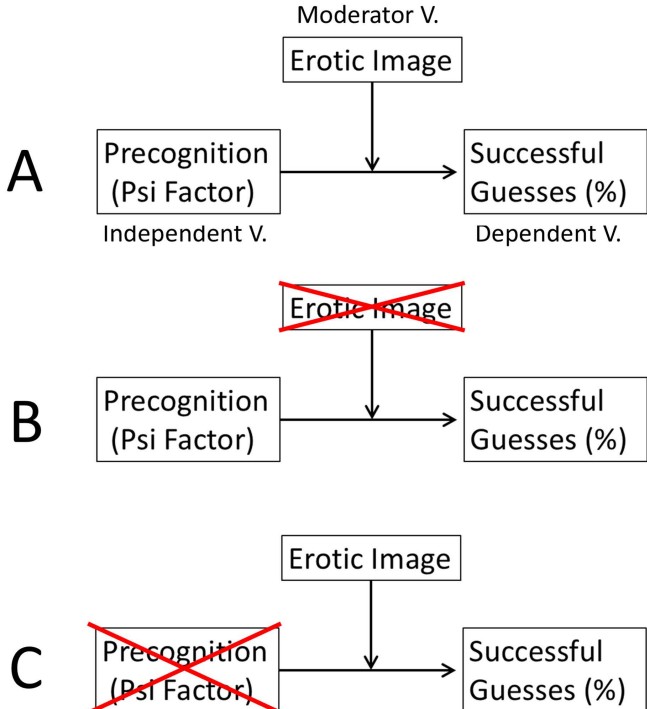

**Fig 1. Overview of experimental and control test strategies. (A)** The basic research paradigm that was used in original Bem Experiment 1 [35] and the TPP Replication Study [20]. The effect of the tested independent variable (precognition/psi factor) on the dependent variable (% successful guesses) depends upon the presence of the (dichotomous) moderator variable (the future display of the erotic image as a reward for guessing successfully). **(B)** CFME-based control strategy that eliminates the use of the moderator variable, i.e., the erotic image that delivers the reward for guessing successfully, from the scientific process. **(C)** CFME-based control strategy that eliminates the independent variable, i.e., the psi factor or psi ability (precognition), from the scientific process. V. = Variable.

First, Fig 1A illustrates the experimental paradigm of Bem Experiment 1 [35]. In this setup, the assumed precognitive ability of participants is the independent variable. The dependent variable was defined as the percentage of correct predictions regarding the side (left or right) of a computer screen on which an image will occur (i.e., the successful guess rate). According to Bem's interpretation of his observations, the nature of the image turned out to be a crucial moderating variable: only for erotic images the percentage of successful guesses differed significantly from mean chance expectancy [35]. No statistically significant deviation was, however, reported for trials that used non-erotic (neutral) images. Therefore, Bem [35] assumed that the observation of null results with the non-erotic (neutral) images served as an important control validating the used research design; however, for an early criticism of that assumption see Wagenmakers et al. [16].

Second, Fig 1B illustrates the CFME-based control strategy that eliminates the display of the erotic image from the used experimental process. That is, the moderator variable, i.e., the future reward experience using erotic images, is substituted for by so-called 'sham-erotic' (image) trials. During such trials, the reward image is not displayed on the computer screen even if the participant has successfully guessed the correct target side. More specifically, in the case of the sham-image trials, only a grey background image is revealed to the participant—irrespective of whether the guess by the participant was successful or not. In AMP terminology, this represents a type of (counterfactual) sham-experimental condition for detecting possible bias in association with the use of the moderator variable. For more explanations and an additional illustration see Fig 3 in Section 2.2.2.

Third, Fig 1C illustrates the CFME-based control strategy that removes the independent variable from the experimental process. This complies with the standard definition of a control condition, namely the assessment of the dependent variable in absence of the independent variable. As described before, in the present study, the independent variable is the psi-anomalous factor (i.e., precognitive ability) associated with the study participant. Consequently, the CFME-based design of this replication project included a comparative test strategy involving both experiments performed in the presence as well as in the absence of participants (using both erotic and non-erotic images). Specifically, for the participant-absent experiments, a random event generator (REG) substituted for the human operator in the performance of the "guessing" task (see Fig 2). Here, the counterfactual (control) condition seeks an answer to the question: Can humans outperform the performance of a REG in predicting the correct target side on the computer screen? Such a control condition had neither been implemented in original Bem Experiment 1 [35] nor in subsequent replication experiments (e.g., [36]).

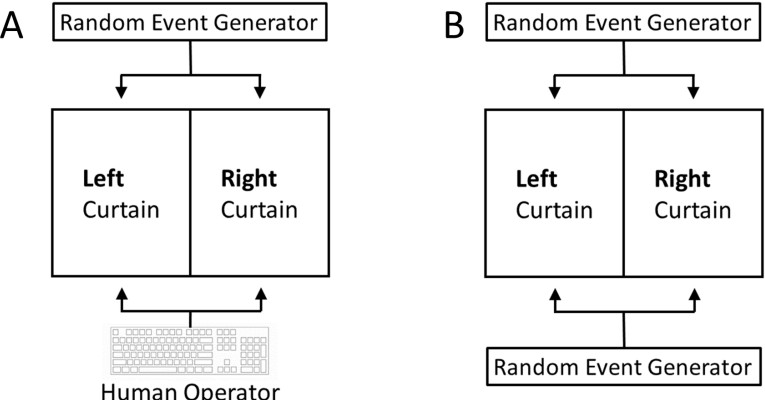

**Fig 2. Illustration of the comparative CFME-based test strategy using a REG. (A)** In Bem Experiment 1, the source of the independent variable is the human operator (the participant) who presses one of two keys (A or L) on a keyboard—indicating the guess of the target side (the left or right curtain behind which the image is presumed to be hidden) on a computer screen. The target side and the type of target image (erotic or non-erotic) is determined using a REG. **(B)** In experiments performed in the absence of participants, an additional, second REG substitutes for the human operator in choosing left or right.

It can be safely assumed that a REG does not possess the tested psi ability, i.e., human precognition. Therefore, the test condition where a REG replaces the human operator (Fig 2B) qualifies as a true control condition—in the sense that the tested independent variable is absent during data collection with either erotic or non-erotic images. Using CFME, it can be empirically assessed whether there is systematic bias in the employed scientific process. This assessment includes the detection of possible bias for outcome measures that solely rely on tests against the theoretical ideal of chance (e.g., see the unknown presence of a biased coin in coin flipping experiments). Again, such forms of bias are referred to as class-A error in AMP terminology [25]. In summary, two different CFME-based test strategies served as advanced quality controls—in addition to the TPP procedures [20]—for validating the reliability of the scientific process as implemented in this AMP-TPP replication study of Bem Experiment 1 (Figs 1B and 1C).

### 1.5. On the use of post-confirmatory hypothesizing in psi research

For this replication project, it was decided to report exploratory analyses also. The reason for this decision was the recognition that exploratory analyses of data sets collected originally with a preregistered confirmatory protocol have also become common in psi research. This new (questionable) research practice is here referred to as post-confirmatory hypothesizing (PCH) which, typically, follows three steps: (i) The investigator finds that a confirmatory study yielded null results with the pre-specified or preregistered analysis for hypothesis testing, (ii) the investigator next comes up with one, or even several, post-hoc developed hypotheses and tests them on the already collected data, or selected subsets of data, with new data-analytic routines, and (iii) upon coming across a new (exploratory) hypothesis that produces a significant result after having re-analysed one of the confirmatory data sets, this result is reported by the investigator as evidence for a new effect. For one example where—using PCH—an unsuccessful confirmatory psi study was turned into an apparently successful psi study see ref. [22]. However, again, there is no confirmatory validity of exploratory psi findings. Such unconfirmed (exploratory) results often distract from the confirmed (null) results.

Therefore, it was decided to report PCH-based findings only if the post-hoc hypothesis would be subject to a subsequent, confirmatory test using a preregistered protocol. In this way, it was assured that both exploratory findings as well as the results of additional confirmatory experiments would be part of the same publication. The commitment to this rigorous research and publication strategy ensures that the credibility of the present research project is maintained despite engaging in PCH-based practices also. Finally, the next section concludes the introduction with a statement concerning (i) the integration of the different studies into a single project for publication, and (ii) the generalizability of the here-presented rigour-enhancing techniques and experimental strategies.

### 1.6. The AMP-TPP metascientific replication project

The present project performed three replication studies, whereby the number of trials increased with each consecutive study (Study 1: N = 37,836 trials; Study 2: N = 127,000 trials; Study 3: N = 217,800 trials; see Section 2.6.9. for the rationale of sample size determination). Normally, the three replication studies might be published separately because each one—on its own—represents a complete effort using a high-powered confirmatory design. Instead, to facilitate comprehension of the evolving nature, including replicability or non-replicability, of the results, the three consecutive studies are combined into a single project for publication. Here, metascientific control techniques for capturing diagnostic information about possible confounding influences are described, including the results from meta-experiments where a REG substitutes for the participants. This allows for the assessment of the integrity of the employed scientific process based on multiple replication studies over time, including with exploratory and confirmatory types of analyses. In summary, this scientific investigation of the scientific process—as used in pursuit of a claimed psi effect—yields new insights into (i) the replicability of an apparent psi-anomalous result and (ii) how advanced control test strategies can help improve the credibility, not only of psi research, but any type of weak-effects experiments probing the frontiers of science.

## 2. Methods

### 2.1. Ethics committee approval and preregistration

The three studies performed with participants were approved by the ethics committee of the Institute for Frontier Areas of Psychology and Mental Health (IGGP), Freiburg, Germany (https://www.igpp.de). The ethics committee was chaired by Dr. Jürgen Kornmeier. The project approval number for the IGGP institutional review of the use of human subjects in this project is IGGP_2020_02. Together with the counterfactual experiments performed in the absence of participants, these studies were preregistered in four preregistrations at the online platform Open Science Framework (OSF): Study 1 with participants: https://osf.io/s9bqr. Study 2 with participants: https://osf.io/3wqk8. Study 1 and Study 2 without participants: https://osf.io/3gtqk. (Note that this preregistered project for the studies without participants was titled AMP-TPP-3 study in the original preregistration document). Study 3 with and without participants: https://osf.io/qkv6p. (Note that this last preregistered project was titled AMP-TPP-4 study in the original preregistration document). For easy access to all primary data and analysis codes for the AMP-TPP metascientific replication project, including access to Github repositories, the following OSF repository was created: https://osf.io/8jteq/. There, all data (https://osf.io/68wy2/files) and analysis codes (https://osf.io/k5fnx/files) can be viewed.

### 2.2. Experimental procedures

This study's main objectives were (i) to establish a new methodological standard for frontiers research and (ii) by applying this standard to confirm or disconfirm—with a high degree of reliability—the replicability of the precognitive effect claimed by Bem Experiment 1 [35]. To this end, multiple replication studies were performed, and the used procedures closely reproduced the TPP procedures [20].

#### 2.2.1. The TPP procedures.
The TPP was developed using a consensus-design process that included Bem himself [37], to conduct a conceptual replication of Experiment 1 of Bem [35] under—up to this point—highest standards of transparency. For details see Kekecs et al. [20]. There, the experimental procedures are described as follows: "The experimental paradigm closely matched the protocol of Bem (2011) Experiment 1. After receiving a briefing about the experiment and its goals from the experimenter, participants completed 36 trials in a laboratory setting, in each of which they were presented with two curtains on the computer screen and had to guess which one hides a picture. If the participant chose the correct (target) curtain, the 'reward image' was revealed, otherwise, a uniform grey background was revealed. Importantly, the target side (left or right) was determined randomly by the computer after the participant's guess. This randomization was done with replacement, so in each trial, the target side was completely random. The reward image for each trial was also determined after the participant's guess by randomly selecting from a pool of 36 reward images without replacement. The pool contained 18 erotic and 18 non-erotic images, resulting in 18 'erotic trials' and 18 'non-erotic trials'. The randomization of the target side and the reward images were independent of each other. The outcome of the trial was whether the participant successfully guessed the target side. The erotic images were selected from the erotic subset of the NAPS image set, while the non-erotic images were from the IAPS. Participants also completed two brief questionnaires: one about belief in and experiences with ESP, the other one related to sensation-seeking. We included these questionnaires to match the original protocol by Bem as closely as possible, because it is unclear whether exposure to these questionnaires would alter the outcome of the study. However, data from these questionnaires are not used in hypothesis testing. Just like in the original experiment, an image of the starry sky appeared, and participants got relaxation instructions before the experimental trials began: 'For the next 3 min, we would like you to relax, watch the screen, and try to clear your mind of all other thoughts and images.' The same image appeared between each trial for 3 s, and participants were instructed to 'clear their minds for the next trial' during this time" (p 5; [20]).

The only substantial differences between the present AMP-TPP study and the TPP study described in Kekecs et al. [20] were the following: In the present study, participants completed the trials online on laptops and desktops via the internet

instead of in a laboratory setting. Only minor technical adjustments were made to the software used by Kekecs et al. [20] to allow the online implementation, but the user interface was kept identical. As a consequence of the online setting, there was no personal briefing of participants or Q&A before or after the session. All instructions were instead given in written form via the experimental software. Participants were given the option to contact experimenters via email after the experiment. In short, the changes made to the software code—as originally used by Kekecs et al. [20]—were exclusively a function of adapting its use to online settings where, unlike in university (laboratory) settings, no experimenters were present (for access to the used software code follow the online links below). Additionally, compared to the prior TPP study [20], another major difference was that the present AMP-TPP study implemented key elements of the AMP methodology [25] for revealing potential hidden biases that might be intrinsic to the employed scientific process when using the TPP procedures. The code for the experimental software used in the AMP-TPP study can be found online at https://github.com/amp-tpp/transparent-psi, https://github.com/amp2-tpp/transparent-psi and https://github.com/amp4-tpp/transparent-psi.

**2.2.2. The AMP diagnostic tool for quantifying systematic errors.** The AMP-based test strategy distinguishes between so-called 'true-experiments' and 'sham-experiments' [25]. For explanation, in the context of the present investigation, true-experiments are called the experiments that are performed by the participants using either erotic or neutral images as moderating variables. By contrast, in the present study, sham-experiments (or sham-test conditions) are the experiments (or test conditions) that are conducted in one of the following counterfactual ways: (i) By the participants in the absence of a moderating variable (see Fig 1B), (ii) by a REG-based process which substitutes for the presence of the independent variable, i.e., the performance of the participant (see Figs 1C and 2B), and (iii) by a REG-based process in the absence of a moderating variable. For more detailed explanations see further below.

For general background, briefly, the choice of how to implement the AMP test strategy in a study depends on (i) the characteristics of the study, and (ii) the decision regarding which systematic error sources should be tracked and quantified. As was explained before in Section 1.3.2., with the AMP diagnostic strategy [25], multiple systematic error types, which are called systematic class-A, class-B, and class-C error in AMP terminology, can be measured. The present replication project focused on the possible presence of class-A error, which presents bias in association with the overall scientific process. For that purpose, as was illustrated in Section 1.4., a counterfactual meta-experimental test strategy was implemented with the AMP [25].

The terminology used in this AMP-TPP replication project, including the abbreviations for the different types of experiments (e.g., X, O, and S), is described next. The experimental condition (X) presents the condition where the participants perform the experiment with erotic images as the moderating variable. The control condition (O) presents the condition where the participants perform the experiment with neutral images as the moderating variable. Again, Bem (2011) reported that experiments with neutral images do not yield the claimed precognition effect; hence, these experiments with neutral images serve as a form of control experiment (O).

As explained above, one type of (counterfactual) sham-experiment removed the independent variable, i.e., the psi factor associated with the participants, from the scientific process while leaving all other aspects of the study as identical as possible (Figs 1C and 2B). Such REG-based sham-experiments, i.e., the experiments without participants, were systematically conducted for both the experimental condition (X) and the control condition (O), whereby—for easy distinction—the REG-based version of the experimental condition (X) is labelled $X_S$ and the REG-based version of the control condition (O) is labelled $O_S$. For example, if such REG-based (counterfactual) experiments, i.e., $X_S$ and $O_S$, were to detect significant deviations from chance, i.e., positive effects, then this would be evidence for the presence of a systematic class-A error and reveal that the used scientific process yields false-positive effects. Again, a "false-positive effect" is an experimental effect on an outcome measure (i.e., the dependent variable) that is falsely attributed to being caused by the tested intervention.

As was already illustrated in Section 1.4., a second type of sham-test condition (S) was performed in Study 1, namely a test condition that removed the moderator variable (i.e., either erotic or neutral images)—but not the participant—from

the experimental process (compare Fig 1B). This meant that so-called sham-image trials (S(X) and S(O)) were performed. As is illustrated in Fig 3B, the so-called 'mixed session' of Study 1 randomly alternated between true-image trials (X and O) and sham-image trials (S(X) and S(O)) such that 18 of the 36 trials in each session were sham-image trials. These sham-image trials served as a counterfactual test condition for investigating the possible presence of class-A error as a function of the moderator variables in the experiment. For an illustration of the differences between mixed-session and pure-session experiments consult Fig 3.

Note that sham-erotic (S(X)) and sham-neutral (S(O)) image trials are identical, except with respect to the labelling of the data—unbeknownst to the participant—as sham-erotic or sham-neutral image trials. Again, when performing these sham-image trials, the participants received no reward; that is, only a grey background was displayed even though the participant had chosen the correct target side. Therefore, the participants thought that they had chosen incorrectly, even if they had been successful in choosing the correct target side as selected by the REG. In short, compared to the pure-session experiment (Fig 3A), the participants in the mixed-session experiment (Fig 3B) unavoidably experienced—on average—less positive feedback, i.e., the experience of a reward in the form of an erotic image upon guessing successfully. This constitutes a difference to the experimental designs implemented by Bem [35] and Kekecs et al. [20]. To be sure, note that the here-employed AMP-based experiments with sham-image trials only served the purpose of revealing possible systematic errors in association with the presence of the moderating variable as part of the scientific process.

Again, the original publication of Bem Experiment 1 [35] claimed evidence showing that only an erotic, but not a neutral (non-erotic), image can act as an effective moderator variable (compare Fig 1A). Therefore, this replication project (just like the TPP) also included experiments with non-erotic images, i.e., 'neutral-image' trials, as a type of control condition (see O in Fig 3). However, for Study 1, the analysis of data from the neutral-image trials was not part of the confirmatory analysis plan to mirror the original analysis procedures as used in the TPP [20]. By contrast, in Studies 2 and 3, the analysis of data from neutral-image trials was also preregistered as part of the confirmatory analysis plan (see Section 2.6.6.).

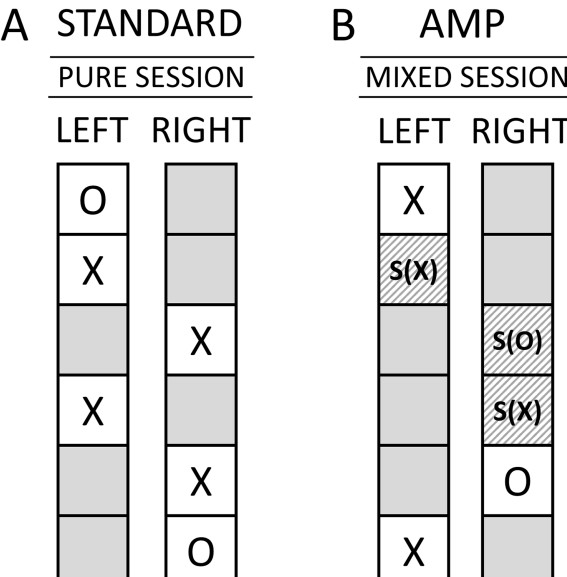

**Fig 3. Illustrative random sequences are depicted for the two types of sessions in Study 1. (A)** Standard experiment with the so-called 'pure session'. Here, each participant performed 36 true-image trials with erotic (X) or neutral (O) images. **(B)** AMP-based experiments with the so-called 'mixed session'. Here, each participant performed 18 true-image trials with erotic (X) or neutral (O) images, and 18 sham-image trials which are labelled S(X) or S(O). Left and right indicate the two sides of the computer screen (compare Fig 2). For more details see main text.

Finally, a third type of sham-test condition (S) was performed in Study 1, namely a test condition that removed both the moderator variable as well as the participant from the experimental process; again, here, REG-based (sham-)experiments were performed in the absence of the moderating variable. The sham-image trials that were performed by the REG are labelled $S(X)_S$ in order to clearly distinguish them from the sham-image trials that were performed by participants which, again, are labelled S(X).

## 2.3. Participants and informed consent

Participants were recruited by an online participant recruitment company (Bilendi), whereby the recruitment period started on December 21, 2020, and ended on November 4, 2022. Informed consent in written form was required for participation. The age of the participants was ensured by the participant recruitment company and confirmed by self-report. The consent procedure was performed by participants presented on the computer screen with extensive information about the study and a consent form with the option to click a button for confirming either agreement or non-agreement. This information was stored in the fully anonymized raw data files as detailed in the data availability statement. Only in the case of agreement with the consent form could participants continue the experiment, otherwise they were automatically logged off by the experimental software. Specifically, eligibility was limited to persons who, according to their self-report, (i) were at least 18 years old, (ii) were not under the influence of alcohol or drugs during the study session, and (iii) did not have a psychiatric illness. Individuals diagnosed with psychiatric conditions were excluded because it was suspected that they may be more likely to have an adverse reaction to the stimuli used in this study. In short, the eligibility check was monitored by the recruitment company as well as experimental software based on self-reporting.

A total of 26,483 participants took part in the study. Participant demographics were as follows: Participants were 40.2% female and 59.8% male, aged on average 49.2 years with a standard deviation of 15.8 years. For compensation, participants received vouchers worth approximately 0.1 EUR/min for their participation, with the participants typically taking between 10 and 20 min to complete an experimental session.

## 2.4. Methods for demonstrating and monitoring protocol fidelity

The present study also employed several methodological tools used in Kekecs et al. [20] as safeguards against possible biases and improving transparency to ensure and demonstrate protocol fidelity. This included direct data deposition, tamper-evident software, pilot studies, preregistration, and open materials. See Kekecs et al. [20] for more details on these methods. In contrast to Kekecs et al. [20], the present study did not employ born-open data, real-time research reports, external research audits and registered reports nor (due to the online nature of the study) laboratory logs, manuals, checklists, and video-verified training.

## 2.5. Randomization

As in the TPP, the Alea algorithm, a state-of-the-art pseudorandom number generator (pRNG) passing the BigCrush test [39], was used for all randomization steps (see https://github.com/nquinlan/better-random-numbers-for-javascript-mirror for more information on the algorithm).

## 2.6. Statistical analysis

The overall aim of the statistical analyses applied in the present study was to confirm, or disconfirm, the hypotheses that participants (or REGs) were able to successfully guess the correct target side with significantly higher (or significantly lower) than chance accuracy in the experimental condition, i.e., the true-erotic trials, but not in any of the control conditions, i.e., the true-neutral, sham-erotic, and sham-neutral trials.

Before describing the statistical analyses in detail in the next sections, it can be noted that they overall followed the analyses of the TPP as outlined in Kekecs et al. [20] except for the following differences: First, for Study 1, only the

primary analyses defined in the TPP were preregistered, not the robustness tests or the exploratory analyses of the TPP. Second, for Study 1 an additional exploratory analysis was conducted, which consisted of a bidirectional variant of the preregistered analysis with a lower significance threshold, applied not only to erotic but also neutral trials (for details see Section 2.6.5.). Third, for Studies 2 and 3, only the exploratory analysis as developed for Study 1 was preregistered as a confirmatory analysis. Next, the various types of data analyses carried out for Studies 1–3 are described.

**2.6.1. Inclusion and exclusion rules.** Incomplete sessions (< 36 trials) were included, to prevent optional stopping bias. Probably due to latency issues of the internet connection, on rare occasions sessions over-ran up to a maximum of 42 trials. This resulted in a very small proportion (< 0.05%) of unplanned extra trials. These trials were also included in the analysis. All trials up to the pre-specified sample size were included in the analysis (Study 1: N = 37,836 trials; Study 2: N = 127,000 trials; Study 3: N = 217,800 trials). All data in excess of the pre-specified sample size was excluded.

**2.6.2. Preregistered statistical analysis in Study 1.** Following Kekecs et al. [20], four statistical tests were simultaneously used to contrast two hypothesized models, namely $M_0$ (assuming no effect) and $M_1$ (assuming an effect). These four tests were pre-specified to be carried out at preregistered analysis points when the completed trial number reached 37,836 (minimum sample size), 62,388, or 86,958 (maximum sample size). The preregistered analysis was to be applied to both true-erotic as well as sham-erotic trials. As specified in the preregistration, the study was to be stopped when all four statistical tests conclusively supported the same model. In the actual Study 1, this was the case at the first analysis point of 37,836 trials per experimental condition. The four tests consisted of a frequentist mixed-effects logistic regression and three Bayesian proportion tests using different priors, as described in the following two sections.

**2.6.3. Mixed-logistics regression model.** The present AMP-TPP study employed the same intercept-only mixed logistic regression model using the glmer function in the lme4 package in R as specified by Kekecs et al. [20], followed by computing the Wald confidence interval for successful guesses around the point estimate of the odds ratio of success, and transforming it to a probability using the logit to probability function. If the upper bound of the confidence interval (CIub) was lower than 0.51, this test would support $M_0$. If the test did not support $M_0$, it was planned to check whether the lower bound of the confidence interval (CIlb) was greater than 0.5. If so, this test would support $M_1$. Otherwise, it would be concluded that the tests did not conclusively support either model. A 99.5% confidence interval (CI) was preregistered for the first analysis point, and 99.75%, 99.875% for the second and third to adjust the CI for repeated testing according to the Bonferroni correction.

The results of the preregistered frequentist statistical analyses were then transformed into a different reporting format for ease of presentation. For each test condition was obtained (i) the observed proportion (%) of hits, i.e., trials in which the predicted side matched with the display side chosen randomly by the REG, (ii) the standard error (SE) of the intercept estimate in the linear mixed model (given by glmer() in the lme4 package in R, corresponding approximately to a 68.3%-CI around the estimated—and true observed—probability of successful guesses), and (iii) the p-value obtained from glmer() in the lme4 package in R. This reporting format was chosen because it is common in the field of parapsychology. Because this reporting format is factually equivalent to the way in which the endpoints were described in the preregistration, no change to the conclusions of the study can occur as a result of it. See Section 2.6.8. for a summary statement clarifying why—and how—deviations from the preregistered analysis were conducted.

**2.6.4. Bayesian proportion test.** Following the procedure outlined in Kekecs et al. [20], the Bayes factor (BF01) was computed based on the total number of successful guesses during erotic trials, and the total number of erotic trials performed by all participants combined, to show the change in the odds of the observed proportion of successes under $M_0$ compared with $M_1$. $M_0$ assumed a success rate of 50% in the population, while $M_1$ assumed a success rate higher than 50%. If Bayes factor (BF01) is lower than 0.04 (1/25), the test would support $M_1$, while if it were higher than 25, the test would support $M_0$. Otherwise, it would be concluded that the tests did not conclusively support either model. To make the statistical inference robust to different analytical decisions regarding the priors, three Bayes factors (BF01) were computed

based on three different prior assumptions about the probability of successful guesses under $M_1$, denoted by three different beta priors, following the identical procedures as Kekecs et al. [20]. There, the three different beta priors were described as the following:

Uniform prior: In this calculation, a beta prior was used with the following parameters: alpha = 1 and beta = 1.

Knowledge-based prior: This knowledge-based prior was originally proposed by Bem, Utts, and Johnson [40], and, thus, it is referred to as the BUJ prior. The prior was described as having a normal distribution with a mean at d = 0 (Cohen's d effect size) and the 90th percentile at d = 0.5. To match this prior in the binomial framework, we used a beta distribution with alpha = 7 and beta = 7. This distribution has 90% of its probability mass distributed between 0.500 and 0.712, where p = 0.712 is equivalent to d = 0.5 effect size. (The formula 'logodds = d × pi/√3' was used to convert d to log odds ratio. Then, log odds ratio was converted to probability using the inverse-logit formula: 'p = exp(logodds)/(1 + exp(logodds))'. Thus, the final equation for getting p from d was: 'p = exp(d × pi/√3)/(1 + exp(d × pi/√3))').

Replication prior: A replication Bayes factor (BF01) was computed where the prior was a beta distribution with alpha = 829 and beta = 733. These parameter values were based on data gathered in Bem's original Experiment 1: 53.1% success rate in 1,560 erotic trials, meaning 828 successes and 732 failures.

**2.6.5. Exploratory (post-hoc) statistical analysis in Study 1.** After analyzing the data collected in Study 1 according to the preregistered confirmatory analysis, a PCH-based exploratory analysis was also performed (see Section 1.5.). This investigation was motivated by the observation of a substantial deviation in the observed hit rate—compared to that expected on average by chance—in the direction opposite to the preregistered hypothesis. The goal of this exploratory analysis was to assess what the inference would have been if using standard statistical approaches currently used in psychological science. The exploratory analysis thus omitted the Bayesian analysis and instead used only a mixed-logistic regression model with a bidirectional variant of the preregistered confirmatory test. Thus, the hypothesis of the existence of a precognitive effect would be considered confirmed if the successful guess rate expected by chance (50% hit rate) lies outside of a 95%-CI of the probability of successful guesses estimated from the observed data using the mixed-effects logistic regression model used in Study 1. For the calculation of p-values consult Section 2.6.3. Note that because of the bidirectional nature of the analysis, this definition of hypothesis confirmation is independent of the direction of the observed effect. The reason for choosing a bidirectional hypothesis—rather than a unidirectional hypothesis in the opposite direction—is that only within the framework of a bidirectional analysis could the observed results of Study 1 be interpreted as a replication of some kind of precognitive effect [35]. The precognitive effect would then be understood as an effect where the participants could both increase as well as decrease (e.g., in the form of a precognitive avoidance effect; see Section 4.3.) the frequency of erotic images being displayed to the participants. In contrast to the preregistered analysis, the exploratory analysis was applied not only to erotic and sham-erotic, but also to neutral and sham-neutral trials (Fig 3).

**2.6.6. Preregistered statistical analysis in Study 2 and Study 3.** In two subsequent replications (Studies 2 and 3), it was explored whether or not the result of the exploratory analysis of Study 1 could be confirmed. To this end, Studies 2 and 3 tested—in a prospective confirmatory fashion—the hypothesis that a significantly higher or lower rate of successful guesses is observed than what is to be expected by chance. For this purpose, the same statistical test as mentioned above in Section 2.6.5. was preregistered to be applied to erotic as well as non-erotic, i.e., neutral, trials, as performed by human participants or a REG. Specifically, Studies 2 and 3 employed an intercept-only mixed logistic regression model using the glmer function in the lme4 package in R as specified by Kekecs et al. [20], followed by computing the Wald-CI for successful guesses around the point estimate of the odds ratio of success, and transforming it to a probability using the logit to probability function. Based on this, a bidirectional analysis was conducted employing a 95%-CI; thus, the hypothesis of the existence of a precognitive effect would be considered confirmed if the successful guess rate expected by chance (50% hit rate) lies outside of a 95%-CI of the probability of successful guesses deduced from the observed data. For the calculation of p-values consult Section 2.6.3.

**2.6.7. Additional preregistered hypotheses for Study 2 and Study 3.** Additionally, a preregistered hypothesis tested whether there was any bias in the left- and right-side choices made by the REG. For this purpose, the 95%-CI around the estimated probability of left-side predictions and left-side targets was calculated in each of the four used test conditions and it was determined whether this CI contained the theoretically expected 0.5 probability of a random left-side choice.

**2.6.8. Deviations from the preregistered analyses.** A substantial deviation from the preregistered analysis occurred in Study 1 when a post-hoc exploratory analysis was conducted even though such exploratory analyses were explicitly excluded as a possibility in the preregistration. This deviation from the preregistration was deemed inevitable based on the realization that—due to the public accessibility of the raw data—post-hoc exploratory analyses would very likely be conducted anyhow by other scientists. It needs to be kept in mind, though, that any exploratory analysis will test a hypothesis different from the preregistered hypothesis. Therefore, no matter what the results of the exploratory analysis are, the conclusions of Study 1 do not change because they are to be purely based on the preregistered analysis for Study 1.

Another difference from the preregistered analysis is the way some of the results are reported in this publication. This difference does not constitute, however, a deviation from the preregistration in any substantial sense (e.g., [41]). That is, the statistical analysis method is not changed but only the results or endpoints are displayed in a transformed way. For each test condition, the following descriptive parameters of the results are reported: (i) the observed proportion (%) of hits, i.e., trials in which the predicted side matched with the display side chosen by the REG, and (ii) the SE and the statistical significance measure p. As mentioned in Section 2.6.3., this reporting format was chosen because it is the most common in the field of psychology and parapsychology. It differs from the form in which results of analyses are described in the preregistration, where the endpoint of statistical analysis was defined in terms of whether the upper or lower boundaries of 99.5%- or 95%-CIs around the observed percentage of correct guesses included or excluded certain values (for more details see Section 2.6.3.). Nevertheless, statistical inference is still based on the criteria specified in the preregistration.

Finally, the preregistrations mentioned a number of possible exploratory analyses for determining whether there existed any unusual features in the temporal dynamics of the rate of correct guesses. Initial inspection of the accordingly plotted data did not, however, reveal any consistent patterns or other noteworthy deviations from chance expectancy. Therefore, this line of investigation was not pursued further and—in order not to distract from the main confirmatory findings—the results from these exploratory investigations will not be reported in the present publication.

**2.6.9. Determination of sample sizes and estimation of statistical power.** Overall, the AMP-TPP studies were designed to provide a power of 0.95, with the aim to keep the probability for alpha-error and beta-error balanced at 5%. Specifically, the pre-specified number of trials to be analyzed in Study 1 was adopted from the TPP study as outlined in Kekecs et al. [20]. There, the authors based their power and sample size calculations on an effect size estimate of a correct guess rate of 51%, taking into account that, although Bem's original Experiment 1 yielded 53.1% successful guesses in trials involving erotic images, later meta-analyses indicated a lower effect. Simulations reported in Kekecs et al. [20] showed that—with a sample size of N = 37,836 trials—the probability of the TPP study to correctly reject the null hypothesis (if the successful guess rate is 51% or higher in the population) is 95% for simulations assuming no systematic individual differences between participants, and 90% for simulations assuming large personal differences. The number of trials in Study 2 was pre-specified at N = 127,000 trials based on power calculations for a one-sample binomial test and Monte Carlo simulations, assuming as an effect size a true successful guess rate of 49.48% (as indicated by the data from Study 1), and a desired power of > 0.95. The same procedure and rationale led to the specification of a sample size of N = 217,800 trials in Study 3. Here, an effect size of a true successful guess rate of 49.61% was assumed (as indicated by the weighted mean of the results found in Study 1 (49.48%) and Study 2 (49.65%)) and the desired power was again set at > 0.95.

# 3. Results

This metascientific project collected the results for three separate studies testing the replicability of Bem Experiment 1 [35]. First, Study 1 reports the results of a confirmatory replication attempt of the claimed precognitive effect and then reports the results from an exploratory analysis of the data from Study 1 (Section 3.1.). Second, Study 2 reports the results of a replication attempt of the exploratory findings from Study 1 (Section 3.2.). Finally, Study 3 reports the results of a second replication attempt (Section 3.3.).

## 3.1. Study 1

Study 1 included two participant-based test strategies which were conducted in parallel. After having completed the participant-based experiments (Section 3.1.1.), data were also collected using a counterfactual test strategy where a REG substituted for the participants (Section 3.1.2.). Finally, after having completed the preregistered (one-sided) analysis for Study 1, an exploratory (two-sided) analysis was also performed on the data from that study (Section 3.1.3.).

### 3.1.1. Experiments with participants in Study 1.

The analysis of Study 1 found that the precognitive effect could not be replicated. All of the frequentist analyses failed to reject the null hypothesis and all preregistered Bayesian analyses confirmed model $M_0$ (see also Section 2.6.2.). This was the case for the erotic trials (X) in both the pure-session (Fig 4A) and mixed-session (Fig 4B) experiments. Upon analysing the counterfactual test condition with sham-erotic trials (S(X)), the employed scientific process appeared to be unbiased as a function of the used reward feedback process; that is, a statistical null result was obtained with sham-experimental condition S(X) as shown in Fig 4B. In summary, this confirmatory replication study was unable to replicate the precognitive effect claimed by Bem (2011).

The Bayesian analysis gave the following results: (1) In the sham-experimental condition (S(X)) within the mixed trials (Fig 4B): The Bayes Factor (BF01) with the most conservative prior (BUJ, see Section 2.6.4.) was 40 (indicating that it is 40 times more likely that the observed data is coming from a population in which the null model is correct compared to the alternative model). (2) In the true-experimental condition (X) within the mixed trials (Fig 4B), BF01(with BUJ prior) = 38, and in the pure trials (X) of the standard experiment (Fig 4A), BF01 (with BUJ prior) = 159.

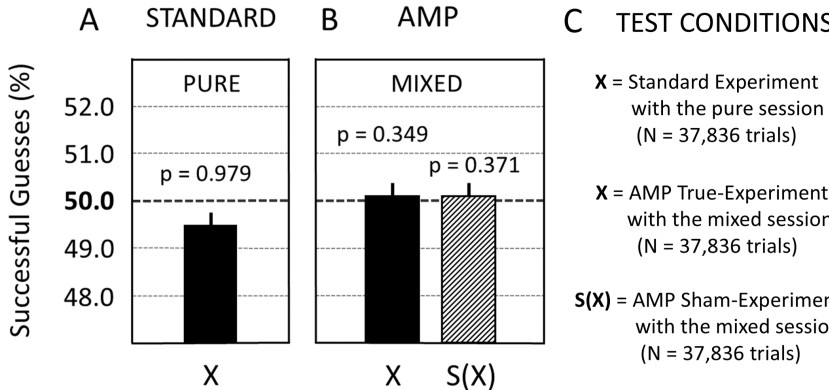

**Fig 4. The results for Study 1 when performed with participants. (A)** Standard experiment with the pure session which reproduced the preregistered, statistical-analytic routines of the TPP: X = 49.48% ± 0.26 (p = 0.979). **(B)** AMP experiments with the mixed session: X = 50.10% ± 0.26 (p = 0.349); S(X) = 50.09% ± 0.26 (p = 0.371). Only erotic trials (X) and sham-erotic trials (S(X)) were analysed as part of the preregistration for Study 1. Whiskers above the bar are standard errors (SE). For details see main text.

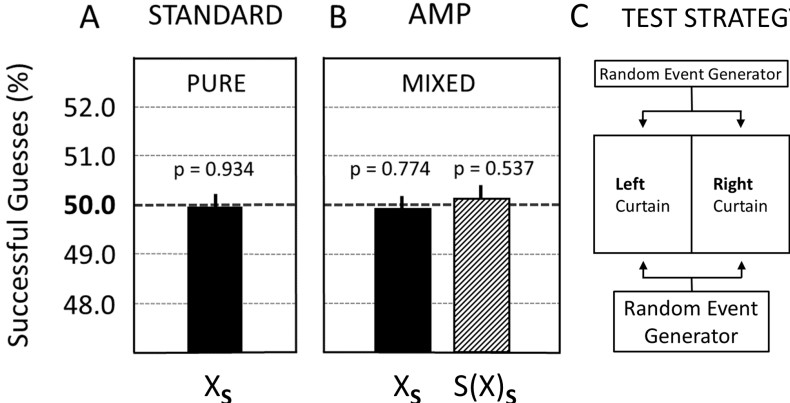

**Fig 5. The results for Study 1 when performed without participants. (A)** Standard experiment with the pure session: $X_S = 49.98\% \pm 0.26$ (p = 0.934), N = 37,826 trials. **(B)** AMP experiments with the mixed session: $X_S = 49.93\% \pm 0.26$ (p = 0.774), N = 37,826 trials; $S(X)_S = 50.16\% \pm 0.26$ (p = 0.537), N = 37,826 trials. **(C)** Illustration of the employed test strategy which substituted a REG for the human operator in the selection of left or right curtains on the computer screen. Whiskers above the bar are standard errors (SE). For details see main text.

The possibility of a more general form of bias was investigated next, i.e., bias in association with the overall scientific process that could have been responsible for the lack of replicability of the precognitive effect for erotic trials (Fig 4A). The question whether such a form of systematic bias (known as class-A error in AMP terminology as was explained in Section 1.3.2.) was present—or not—in the used measurement process was addressed empirically with experiments carried out in the absence of participants (compare Fig 2).

**3.1.2. Experiments without participants in Study 1.** Three confirmatory replication experiments of Study 1 were performed without participants. In these experiments, the REG substituted for the human operator. Only statistical null results were observed for all three REG-based experiments (see Fig 5): For the pure session: $X_S = 49.98\% \pm 0.26$ (p = 0.934); for the mixed session: $X_S = 49.93\% \pm 0.26$ (p = 0.774); $S(X)_S = 50.16\% \pm 0.26$ (p = 0.537). These experiments thus confirmed that systematic measurement biases were unlikely to produce false-negative (or false-positive) results with the frequentist analysis employed in participant-based Study 1, e.g., as a function of hidden bias towards lower-than-chance probabilities of guessing successfully the future random events, thereby validating the reliability of the used research design (see Fig 5). Put differently, it was empirically confirmed—for the here-employed scientific process—that the risk was low of either falsely failing to reject (a false-negative result) or falsely rejecting (a false-positive result) any of the three frequentist null hypotheses for participant-based experiments in Study 1 (Figs 4A and 4B).

The observation of the null results, which were obtained with the preregistered TPP procedures for both participant-based experiments (Fig 4) as well as REG-based experiments (Fig 5), concluded the confirmatory analysis of Study 1. Next, an exploratory (two-sided) analysis of Study 1 was conducted because the visual inspection of Fig 4A suggested an apparent negative deviation for erotic trials (X) in the opposite direction than was predicted by the preregistered, one-sided statistical analysis for Study 1. The size of this deviation was about 0.5% compared to the 50% guess rate predicted by chance.

**3.1.3. Exploratory analysis of Study 1 with participants.** An exploratory analysis of the data collected for confirmatory Study 1 (with participants) was carried out. As was explained in Section 1.5., the practice of performing an exploratory (post-hoc) analysis on data originally collected for the purpose of confirmatory research is here referred to by the term PCH (post-confirmatory hypothesizing). Here, the exploratory analysis consisted of applying a two-sided PCH-based analysis to the observed data (see Section 2.6.5.). The results are summarized in Fig 6.

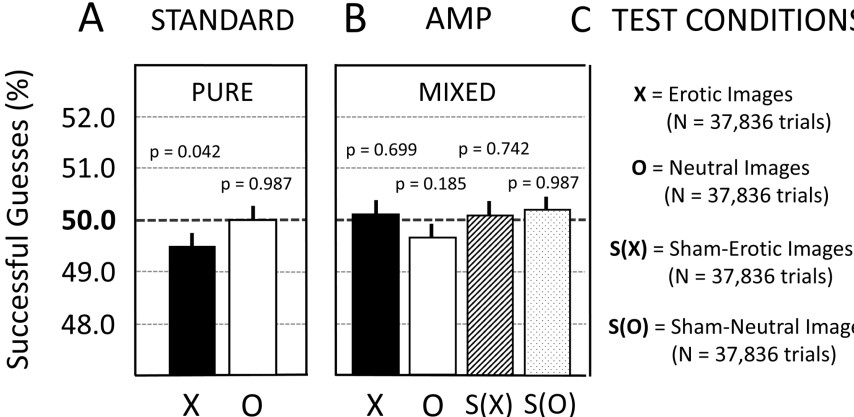

**Fig 6. Exploratory analyses for all six test conditions performed during Study 1. (A)** The results for erotic (X) and neutral trials (O) in the pure session: X = 49.48% ± 0.26 (p = 0.042); O = 50.00% ± 0.26 (p = 0.987). **(B)** The results for the mixed session: X = 50.10% ± 0.26 (p = 0.699); O = 49.66% ± 0.26 (p = 0.185); S(X) = 50.09% ± 0.26 (p = 0.742); S(O) = 50.19% ± 0.26 (p = 0.987). The statistical values are the result of two-sided statistical analyses. For an explanation of the various test conditions see the legend to Fig 3 in the Methods. Whiskers above the bar are standard errors (SE). For details see main text.

The analysis of neutral trials (O) and sham-neutral trials (S(O)) showed statistical null results, both for the pure- and mixed-session experiments (compare Figs 6A and 6B). In addition, as shown in Fig 6B, the mixed-session experiment showed non-significant results for both erotic trials (X) and sham-erotic trials (S(X)). By contrast, the exploratory analysis of the data collected during Study 1 suggested a potential deviation effect of −0.52% for the pure-session experiment (see X in Fig 6A; p = 0.042). Note that a p-value produced by a post-hoc exploratory analysis cannot be safely interpreted to indicate the statistical significance of the associated deviation from mean chance expectancy that was observed in the data. Instead, only p-values derived from confirmatory analyses can be coherently interpreted in terms of statistical significance. Therefore, an apparently significant finding emerging from an exploratory or PCH-based analysis should undergo confirmatory testing before being reported in a publication. To this end, the subsequent Study 2 sought to further investigate this exploratory finding of Study 1 by a high-powered confirmatory replication.

### 3.2. Study 2

Study 2 included a participant-based test strategy (Section 3.2.1.), which was followed by a counterfactual test strategy, where a REG substituted for the human operator in selecting the target side on the computer screen (Section 3.2.2.).

**3.2.1. Experiments with participants in Study 2.** Study 2 performed a confirmatory replication experiment for the exploratory results (for both erotic and neutral trials) from Study 1. See Fig 7 for a comparison of the results between Study 1 (Fig 7A) and Study 2 (Fig 7B).

The preregistered (two-sided) analysis of Study 2 found that the PCH-based results found with Study 1 for erotic (X) and neutral (O) trials (Fig 6A) could be confirmed. As illustrated in Fig 7B, Study 2 replicated the direction of the (exploratory) effect for erotic trials (X), and the percent effect size was on the same order of magnitude as before in Study 1, i.e., X = −0.35% (p = 0.013) in Study 2 (compare X in Figs 7A and 7B). Equally, the statistical null result for neutral trials (O) in Study 1 (Fig 7A) was again observed in Study 2 (Fig 7B), that is, O = 49.83% ± 0.14 (p = 0.236). Note that compared to Study 1, the used sample size in Study 2 was about 3.35-fold larger than in Study 1, i.e., 127,000 trials per test condition (X or O) in Study 2 compared to 37,836 trials per test condition (X or O) in Study 1.

In summary, Study 2 confirmed that an unknown, yet apparently systematic, influence, or factor, produced a significant lower-than-chance probability (below 50%) of guessing successfully; that is, the probability of wrongly guessing the later

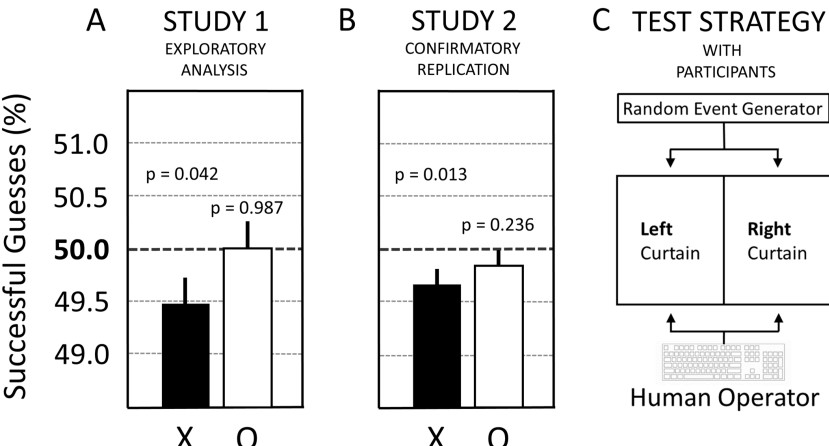

**Fig 7. Result comparison between Study 1 and Study 2 when performed with participants. (A)** Study 1: X = 49.48% ± 0.26 (p = 0.042), N = 37,836 trials; O = 50.00% ± 0.26 (p = 0.987), N = 37,836 trials. **(B)** Study 2: X = 49.65% ± 0.14 (p = 0.013), N = 127,000 trials; O = 49.83% ± 0.14 (p = 0.236), N = 127,000 trials. X = erotic trials; O = neutral trials. **(C)** Illustration of the employed test strategy where a human operator, i.e., the participant, selects either the left, or right, curtain on the computer screen by pressing a button on the keyboard. Whiskers above the bar are standard errors (SE). For details see main text.

computer-generated random events was significant (p = 0.013). Again, this positive result in confirmatory Study 2 agreed with the earlier exploratory finding in Study 1 (compare Figs 7A and 7B).

**3.2.2. Experiments without participants in Study 2.** To assess the risk of the possible influence of systematic error on the used scientific process, the test conditions for Studies 1 and 2 were reproduced as closely as possible—except for the fact that a REG substituted for the participants. The results of these REG-based experiments are shown in Fig 8. Note that the null result for test condition X in Study 1 from Fig 5A is illustrated again in Fig 8A for easy comparison.

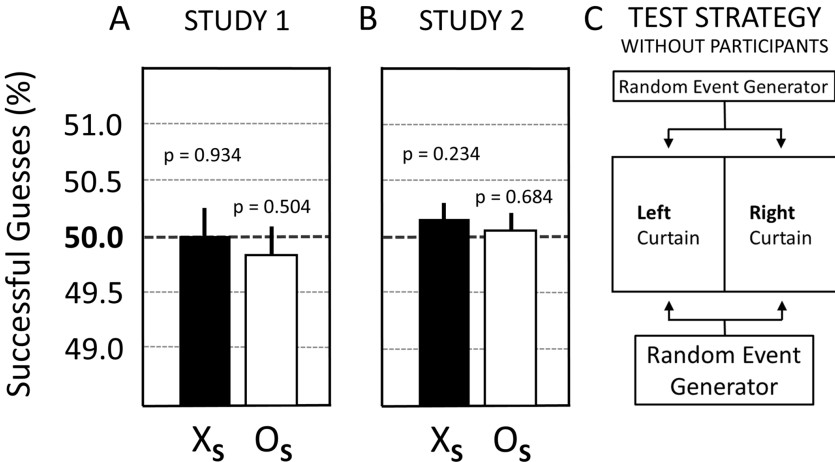

**Fig 8. Results comparison between Study 1 and Study 2 when performed without participants. (A)** Study 1: $X_S$ = 49.98% ± 0.26% (p = 0.934), N = 37,836 trials; $O_S$ = 49.83% ± 0.26% (p = 0.504), N = 37,836 trials. Note that the result for erotic trials ($X_S$) in Study 1 was taken from Fig 5A for easy comparison. **(B)** Study 2: $X_S$ = 50.17% ± 0.14% (p = 0.234), N = 127,000 trials; $O_S$ = 50.06% ± 0.14% (p = 0.684); N = 127,000 trials. $X_S$ = erotic trials; $O_S$ = neutral trials. **(C)** Illustration of the employed test strategy which substituted a REG for the human operator in the selection of left or right curtains on the computer screen. Whiskers above the bar are standard errors (SE). For details see main text.

Fig 8 presents the results for erotic ($X_S$) and neutral ($O_S$) trials when Study 1 and Study 2 were conducted without participants. These experiments where a REG substituted for the human operator (see Fig 8C) confirmed that intrinsic systematic bias was unlikely to produce false-positive (or false-negative) results in participant-based Studies 1 and 2, e.g., as a function of bias towards lower-than-chance probabilities of guessing successfully the random events (compare Figs 7 and 8); that is, only statistical null results were observed for erotic ($X_S$) and neutral ($O_S$) trials with these (counterfactual) REG-based experiments, thereby validating the reliability of the used research process (Figs 8A and 8B). In summary, it was empirically confirmed that the risk was low of falsely rejecting the null hypothesis for erotic trials during participant-based Study 1 (N = 37,836 trials) and Study 2 (N = 127,000 trials) with the here-employed scientific process. The credibility of this confirmatory result is enhanced by the finding that for all other six test conditions in Study 2, i.e., for the conditions predicting only null results (see O in Fig 7 and the REG-based trials in Fig 8), only statistical null results were—in fact— observed. Next, Study 3 sought to determine whether this anomalous result for erotic trials could be replicated a second time when using a larger sample size and a more stringent counterfactual (control) research design.

### 3.3. Study 3

Study 3 conducted a second replication experiment of the results from the exploratory analysis of Study 1 (see Fig 7A) as well as the confirmatory replication experiment in Study 2 (see Fig 7B). Specifically, Study 3 investigated whether repro-ducing—as closely as possible—the test conditions of participant-based Study 2 could—again—replicate the 0.35%-effect (p = 0.013). Study 3 represented a confirmatory replication study based on 217,800 trials per test condition (X or O).

Importantly, Study 3 also included a counterfactual test strategy where a REG substituted for the human operator in selecting the target side on the computer screen. However, in the case of Study 3, the data for REG-based experiments were not collected later—as in the case of Studies 1 and 2—but rather the REG-based experiments and the experiments with the human operators were run in parallel during the same time. The REG-based experiments also collected 217,800 trials per test condition ($X_S$ or $O_S$). See Fig 9 for the results of the participant-based experiments (Fig 9A) in comparison to the parallel REG-based experiments (Fig 9B) for both erotic (X and $X_S$) and neutral (O and $O_S$) image trials.

Study 3 was unable to replicate the prior positive finding from Study 2 (−0.35%; p = 0.013) and only a statistical null result was found for the erotic trials (0.07%; p = 0.496) in Study 3 as shown in Fig 9A. The parallel REG-based experiment

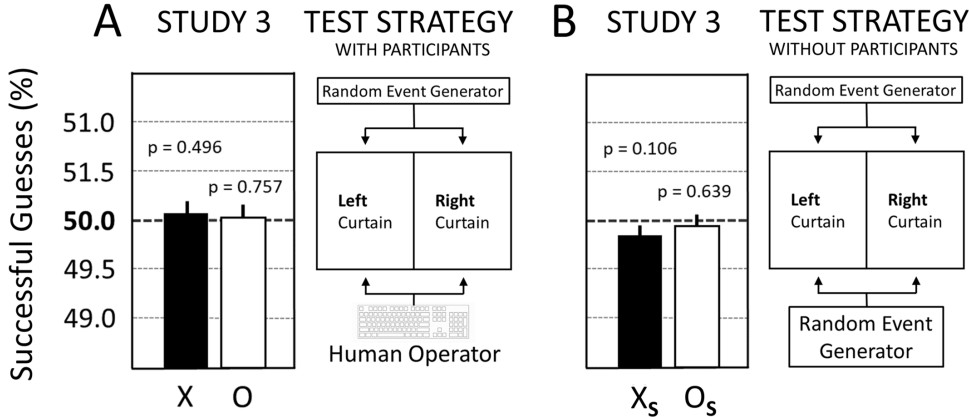

**Fig 9. Results comparison for participant-based and REG-based experiments in Study 3. (A)** Study 3 with participants: X = 50.07% ± 0.11 (p = 0.496), N = 217,800 trials; O = 50.03% ± 0.11 (p = 0.757), N = 217,800 trials. X = erotic trials; O = neutral trials. **(B)** Study 3 without participants (REG-based): $X_S$ = 49.83% ± 0.11 (p = 0.106), N = 217,800; $O_S$ = 49.95% ± 0.11% (p = 0.639), N = 217,800 trials. $X_S$ = erotic trials; $O_S$ = neutral trials. Whiskers above the bar are standard errors (SE). For details see main text.

confirmed the reliability of the used scientific process because in the absence of the source of the independent test variable, i.e., the participant, only the predicted null results were observed (Fig 9B).

### 3.4. Additional preregistered hypothesis testing for Study 2 and Study 3

The preregistered analysis for determining whether there existed any bias in the left- and right-side choices made by the REG (see Section 2.6.7.) confirmed that there was no bias in the randomly determined target side by the computer server (data not shown).

## 4. Discussion

This metascientific replication project performed three confirmatory studies testing the replicability of the precognitive effect claimed by Bem, i.e., the anomalous deviation above 50% in a binomial statistical test (53.1%; p = 0.01) as reported for Experiment 1 [35]. Study 1 reported (i) a failure to replicate Bem's original findings, and no indication for there being a precognitive effect using the confirmatory analysis and (ii) an anomalous deviation from mean chance expectancy in the direction opposite to that predicted originally (49.48%) with an exploratory analysis. It was decided to conduct a new confirmatory study to examine the replicability of this exploratory finding. This Study 2 replicated (49.65%) the exploratory below-chance finding from Study 1. Finally, Study 3 reported an unsuccessful replication attempt (50.07%) of the finding from Study 2. In short, the once-confirmed anomalous result in Study 2 could not be confirmed a second time with Study 3.

Nevertheless, for the field of parapsychology, the results of Study 2 present a rare instance of a successful replication attempt of an exploratory finding using a confirmatory protocol; however, again, the true source of the one-time confirmed anomalous result in Study 2 remains to be identified (see Section 4.3. for more details). In short, none of the three replication studies was able to detect the precognitive effect (53.1%) reported by Bem [35]—neither with exploratory (PCH-based) nor confirmatory analyses. Next, Section 4.1. will summarize key findings that were obtained with the AMP diagnostic test strategy for reducing the risk of false discoveries.

### 4.1. The scientific investigation of the scientific process confirms a reduced risk of bias

In addition to implementing a high-powered replication study design (involving 26,483 participants who performed a total of 420,472 trials with erotic images), this work included a large-scale (meta-)scientific investigation of the reliability of the employed scientific process itself; therefore, again, the description of the present work as a 'meta-scientific replication project'. Using counterfactual test strategies for capturing diagnostic information about the scientific process with the AMP (Fig 1), only the predicted null results were observed when performing a total of 420,472 REG-based (counterfactual) trials, i.e., trials performed in the absence of participants (see Figs 5, 8, and 9B). This presents empirical evidence indicating the absence of intrinsic systematic measurement bias in the experimental system. Consequently, compared to ordinary research designs, which do not actively control for intrinsic bias with diagnostic calibration routines, the likelihood of reaching false-negative or false-positive conclusions was reduced with the here-adopted AMP-TPP test strategy. The use of this advanced methodological procedure is of course not limited to the field of parapsychology but instead can increase reliability of results in all research areas. Next, Section 4.2. will compare the results of this AMP-TPP replication project with the prior results of the TPP project by Kekecs et al. [20] for Bem Experiment 1 [35].

### 4.2. Comparison of the results from the AMP-TPP replication project with Kekecs et al. (2023)

Including the earlier report by Kekecs et al. [20], none of the four replication studies using the TPP procedures was able to report a greater-than-chance probability (above 50%) of successfully guessing future random events; that is, the results claimed by Bem [35] and Bem et al. [36] were not found to be replicable when using the consensus-designed TPP procedures (Table 1).

**Table 1. Results of the four replication studies of Bem Experiment 1 [35] with the TPP procedures.**

| Reference | Result (%) | Analysis | No. Erotic Trials |
|---|---|---|---|
| **TPP Study** | 49.89 (99.75% CI: 49.1, 50.7) | One-sided | 37,836 |
| **AMP-TPP Study 1** | 49.48±0.26; p=0.979 | One-sided | 37,836 |
| **AMP-TPP Study 1ª** | 49.48±0.26; p=0.042 | Two-sided | 37,836 |
| **AMP-TPP Study 2** | 49.65±0.14; p=0.013 | Two-sided | 127,000 |
| **AMP-TPP Study 3** | 50.07±0.11; p=0.496 | Two-sided | 217,800 |
| **Bem (2011)** | 53.1; p=0.01 | One-sided | 1,560 |

The TPP Study is the study by Kekecs et al. [20]. The set of studies labelled AMP-TPP Study 1, AMP-TPP Study 2, and AMP-TPP Study 3, comprises the metascientific project reported in the present article. For easy comparison, the relevant result by Bem [35] is included at the bottom of Table 1. ª Exploratory analysis (two-sided).

Note that for these overall null results it did not matter whether participants were performing the guessing task in a university setting [20] or online, e.g., in the comfort of their home (as in the present work). In summary, with respect to the failure to replicate the precognitive effect claimed by Bem [35], the results of this AMP-TPP replication project fully agree with the prior negative findings of the TPP study by Kekecs et al. [20]. Furthermore, the TPP methodology developed by Kekecs et al. [20] was confirmed to be reliable according to the diagnostic measures of the here-employed metascientific strategy (Section 2.2.2.); that is, empirical evidence was obtained that indicated a low risk of reaching false-negative or false-positive conclusions with the TPP.

In contrast to the TPP Study by Kekecs et al. [20], the AMP-TPP Study 2 found evidence for a significant lower-than-chance probability of successfully guessing the outcome of future random events (49.65% instead of the expected 50.00%; see Table 1). Similarly, the successful guess rate was also below 50% in the AMP-TPP Study 1 (49.48% as shown in Table 1). The source of the anomalous results in Studies 1 and 2 remains to be identified. There are two possibilities: either (i) a psi-derived anomaly was revealed, although a non-replicable one, or (ii) a method-derived anomaly was detected—resulting from a statistical fluke or systematic error which remains to be identified. For deeper discussions regarding these two possibilities, including the limits of the present replication project, see the subsequent sections. In any case, the existence of a consistently replicable anomalous effect was disconfirmed by the statistical null result in AMP-TPP Study 3 (see Table 1).

## 4.3. The anomalous result of (AMP-TPP) Study 2: Psi-based versus conventional interpretation

Interpretations of findings in psi research have often called on considerations from the philosophy of science, including scientific epistemology and questions concerning the foundations of science (e.g., [13,42–47]). This is to be expected for the following reason alone: If the existence of psi-anomalous phenomena could be confirmed, then the nature of the scientific method—as it is currently understood—would need to be revised. For example, this includes the standard scientific assumption that human consciousness cannot affect scientific measurement processes—other than by conventional means. In short, interpretations of scientific results, including possible psi-based interpretations, take place—invariably—using certain epistemological assumptions and world views. While this is usually of minor importance in standard science, the situation is different when it comes to hypotheses that challenge established worldviews. Thus, interpretations of findings in psi research are typically influenced by preferences arising from different philosophical positions regarding the fundamental nature of reality and consciousness.

For example, someone who adopts the metaphysical position of panpsychism might interpret an observed anomaly in experimental parapsychology using a different frame of reference (e.g., psi-based interpretation) than someone who routinely adopts the metaphysical position of physicalism (e.g., conventional interpretation). To account for this situation in the subsequent discussion, a neutral position will be adopted with respect to metaphysical assumptions, and—whenever

necessary—the respective reference frames will be pointed out. In that spirit, the following questions will be addressed: What are important limitations of the present replication project? What important questions remain unanswered?

A psi-based interpretation might argue that statistical-significant results in a psi-research project should right away be interpreted as evidence for some kind of psi effect and, therefore, the following psi-affirmative questions would likely arise: Why was the direction of the psi effect observed in Study 2 reversed compared to the original (precognitive) effect [35]? And why could the psi effect observed in Study 2 not be replicated in Study 3? Regarding the apparent effect reversal in Study 2, i.e., the observation of an anomalous, below-chance rate of successful guesses, this finding could, for example, potentially be indicative of a precognitive avoidance effect, where participants use their precognitive ability to avoid being shown an erotic image. In other words, such a psi-based interpretation might argue that the hypothesized precognitive abil-ity can manifest both as precognitive avoidance and precognitive preference. Furthermore, a psi-based interpretation might propose to account for unstable and non-replicable results by referring to additional unconventional (psi) ideas like the "trickster effect", "psi-missing", or "experimenter psi" (e.g., [13,42–45]). For explanation, these concepts are often referred to in the parapsychological literature for the purpose of explaining the non-replicability of an original finding—using unproven psi-based assumptions. For example, the concept of 'experimenter psi' claims that the psi ability of the experimenter inter-feres with the experiment and—therefore—the true cause of an observed null result in a replication study is the experiment-er's own psi (e.g., [43,44]). No reliable evidence in support of this claim exists. For a critical discussion of such psi-based concepts in relation to the frequent non-replicability of results in psi research see Section 4.7. More generally, some special, not-yet identified, psi-conducive factor could have been missing in the present study and—therefore—the original psi effect of Bem [35] could not be replicated. In summary, the anomalous result of Study 2 could be interpreted as direct evidence for a psi-derived anomaly, although one that could not be confirmed a second time with Study 3.

From the perspective of conventional science, the fact that an initial exploratory observation can be replicated suc-cessfully with a high-powered, confirmatory study would likely be viewed as credible evidence supporting the existence of the hypothesized phenomenon. However, as is required for all forms of anomalies research exploring the frontiers of science, higher standards of evidence are also required for research on anomalous cognition to better reduce the risk of false discoveries. For example, a lower threshold for the acceptable alpha-error rate is often required in such cases before drawing conclusions. The obvious reason is that claims of anomalous findings often challenge long-standing conventional assumptions about the nature of physical reality. For example, in the case of parapsychology, not only do claims of pos-itive results question the limits of psychological science but—critically—the limits of the known laws of physics (see also Section 1.). Note that such challenges to conventional physical theory present themselves also in other cases of anoma-lies research such as, for a recent example, in the search for the anomalous magnetic moment of muons in high-energy physics [48]. For a brief discussion comparing the challenges of anomalous cognition research with those encountered in anomalies research in physics consult Section 4.6.

Given the high stakes outlined above, the integrity of the used research processes is of special importance when inves-tigating anomalies in any field of scientific investigation. However, while diagnostic calibration procedures—for verifying the accuracy of signal detection processes—are routinely adopted, for example, in the search for anomalies in phys-ics, the systematic implementation of calibration procedures, i.e., those dedicated to revealing hidden systematic error sources, is extremely rare in psychology or psi research (e.g., [25]). In the present replication project, the AMP-based approach implemented diagnostic calibration routines such as sham-test conditions (e.g., sham-erotic trials) and counter-factual meta-experiments (i.e., REG-based studies without participants) to assess the integrity of the employed scientific process and to determine if there were hidden biases intrinsic to the research methodology used in the detection of the (anomalous) precognition effect. As was described in Section 1.3., these diagnostic calibration procedures for increasing the reliability and accuracy of the results were used in addition to using preregistered protocols and data transparency because, again, such intrinsic sources of systematic error cannot—necessarily—be revealed, or eliminated, by preregis-tered confirmatory testing or methodological transparency.

In summary, the results presented here can theoretically be interpreted both in terms of a psi-based explanation as well as conventional explanation. The psi-based interpretation suffers from the fact that so far no empirically testable hypotheses have been derived from the ideas concerning psi-based (anomalous) explanations of non-replicability (see also Section 4.7.). Nevertheless, the psi-based proposition of an unknown or uncontrollable psi factor always remains a possibility as an "explanation"—even if only as part of a preferred philosophical position—for an anomalous result that can be confirmed only once. By contrast, the conventional interpretation of the results comes with the benefit of not requiring any additional assumptions, since the results can simply be seen as statistical artefacts, for example, due to conventional sampling error. At the same time, the magnitude of the deviation observed in Study 2 is such that it would be considered—by convention—a rather unlikely event to occur by chance alone. Other possible sources of experimental error or measurement bias are not obviously apparent and have to some extent been controlled for by the counterfactual (e.g., REG-based) test strategies in this metascientific replication project. Therefore, for the time being, the true source of the observed anomalous deviation from chance remains unidentified and, hence, the scientific status of the once-replicated anomalous effect observed in Study 2 will be characterized as an 'unidentified laboratory phenomenon', or ULP in short.

## 4.4. On classifying all existing results in parapsychology as ULPs

For brief review, under the psi hypothesis, the independent variable to be tested with an experimental psi study is the psi factor associated with the participant. However, in the history of parapsychology, none of the published studies was able to identify this psi factor as the true source of an observed anomalous result. Typically, the psi investigator detects an anomalous result in an experimental study and then reports the result. This typical scenario is represented by Step 1 in Fig 10A. However, the second step involving the identification of a psi factor, or psi ability, as the true source of the anomalous result has never been successfully undertaken in psi research (Step 2 in Fig 10B); hence, the above use of the term ULP.

The above-described challenges for parapsychological research must be addressed in the choice of the scientific process that is used for a given experiment, including the rules for data interpretation; so far, these challenges—unfortunately—have not yet been credibly addressed in the field of parapsychology for the findings that have been promoted as evidence for true-positive (psi) effects—without having identified the true source of the anomalous results. Furthermore, if the work of identifying the true source of an anomalous signal is based on an Exclusion Principle only, i.e., the mere exclusion of known sources of error in the study, then no strong claims about the true signal source can be made. Again, this is a particular concern in the absence of (i) replicable findings (as in the present study) and (ii) accepted theories for predicting specific measurement outcomes (see Section 4.5.). In summary, until the time when the true source of an apparent psi-anomalous phenomenon can be identified in a reliable manner, its scientific status is that of an ULP—instead of, prematurely, a psi-derived anomaly, i.e., a true-positive result (Fig 10).

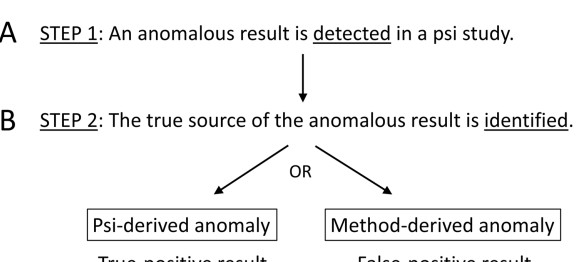

**Fig 10. Anomalies discovery process. (A)** The detection of an anomalous result (Step 1). **(B)** The identification of the true source of the anomalous result (Step 2).

The parapsychological literature has reported many 1,000s of such ULPs over the last 100 years, for a wide range of experimental paradigms—from the Ganzfeld experiment that claimed evidence for anomalous information transfer (telepathy) to REG-based studies testing for psychokinetic abilities (e.g., [28–31]). Obviously, the question is not whether 1,000s of psi studies have reported statistically significant results; instead, the all-critical question is: Why have these studies shown positive results? Was there a psi factor operating in these studies? The crucial question for parapsychology remains: What are the true sources, or origins, of the many significant anomalous results? For only one example, given the mostly exploratory nature of the reported psi studies, the possibility cannot be excluded that the true source of many of the observed significant deviations might simply be statistical sampling error.

Finally, finding credible answers continues to present a particular problem in the field of anomalous cognition for this reason also: Plausible theoretical predictions are lacking that could guide the identification of an anomalous result as a psi-derived anomaly. Concerning the still-unresolved task of predicting outcomes in psi experiments, one type of theoretical effort has long created interest in the field of parapsychology: The attempt to use the principles of quantum mechanics for crossing the divide between mind and matter. A brief historical overview is described next.

## 4.5. On the claim of psi phenomena as macroscopic quantum phenomena

For many decades, parapsychologists have attempted to connect psi phenomena with the field of quantum physics. For example, invoking the quantum interpretation known as the Copenhagen interpretation, it was argued that anomalous mind-over-matter effects, e.g., psychokinetic effects on REG devices, might be explained by an observer-induced collapse of the quantum state vector (e.g., [49,50]). In this way, so it was thought, a connection could be established between a person's conscious intent and a (quantum-random) physical process through the principles of quantum mechanics; again, the hope was that apparent (psi) correlations between observer intent and physical (REG) outcomes could be explained as a quantum phenomenon manifesting at the macroscopic level of laboratory experimentation. In a further development, the concept of quantum entanglement, which underlies the observation of nonlocal correlations in laboratory settings (e.g., [51,52]), has long been considered as an explanation for (claimed) psi abilities such as precognition and remote viewing (e.g., [53]). Briefly, the concept of quantum nonlocality refers to the long-confirmed observation of informational correlations between quantum events occurring at two distant physical locations. Remarkably, these long-distance (above-chance) informational correlations can exist in apparent contradiction to classical space-time constraints (e.g., in violation of the classical assumption of local realism). However, it is well-understood that proposals for explaining intentional, that is, non-random, psi phenomena based on nonlocality and nonlocal correlations are incompatible with the established requirements of quantum theory. The subsequent paragraph will explain this fundamental constraint in more detail.

Quantum mechanics prohibits any degree of (intentional) control via nonlocal (quantum) correlations. Specifically, recognizable macroscopic nonlocal correlations between intentional states and quantum-physical outcomes (e.g., psychokinetic effects on REG devices) are prohibited by the non-signalling principle (also known as the non-communication theorem) of quantum mechanics; otherwise, paradoxical scenarios could manifest in reality because of the possible transmission of faster-than-light signals, messages, or communications—in direct contradiction to relativity theory. For early discussions see Eberhard [54] and Bell [55]. In short, the (complete) quantum state must remain inaccessible to any possible observer/agent for the purpose of controlling nonlocal outcomes in experiments based upon entanglement correlations. This must hold true for any form of quantum interpretation, whether for Ψ-epistemic interpretations, such as the Copenhagen interpretation, or Ψ-ontic interpretations, such as Bohmian quantum mechanics with its nonlocal ontology or quantum mechanics based on time-reversing, retrocausal ontologies (e.g., [56–59]). Note that—in the context of quantum-theoretical interpretations—the terms 'Ψ-epistemic' and 'Ψ-ontic' do not refer to any psi-anomalous forms of cognition (i.e., parapsychology) but merely to the quantum-mechanical wave function Ψ which can be interpreted either epistemically or ontically in the respective interpretations of quantum mechanics (for further explanations see ref. [59]). In summary, the non-signalling principle (i) prohibits accessing and controlling quantum-entangled states in line with an

agent's intentions about creating specific nonlocal outcomes, and, thereby, (ii) guarantees the compatibility of quantum and relativity theories. Finally, again, the non-signalling principle equally applies to any form of time-reversing (retrocausal) quantum interpretation. That is, the non-signalling principle prevents the (quantum) transmission of signals or messages from the future to the present or past (e.g., [56]). This well-known constraint on transmission directly challenges the plausibility of prior work claiming that quantum theory might explain the anomalous phenomenon of precognition (e.g., [60,61]).

In recognition of the limits that are imposed by the non-signalling principle, a further modelling attempt for explaining psi phenomena—based on quantum entanglement—was offered in the form of Weak or Generalized Quantum Theory (GQT; [62,63]). Basically, the GQT-model argues that—while intentional (non-random) psi effects are indeed prohibited by quantum theory—randomly-occurring psi phenomena, such as (non-intentional) spontaneous telepathic transmissions between two persons, would still be compatible with quantum theory [63]. This led to the GQT-guided proposal stating that while laboratory psi effects are indeed non-replicable in principle, the (anomalous) non-replicability of psi effects could—nevertheless—be distinguished empirically from conventional types of non-replicability. This might be achieved by way of systematic, higher-level experimental strategies, e.g., in the form of so-called matrix-correlation experiments (e.g., [64]); however, whether potential positive findings with such GQT-based matrix-correlation experiments would agree with all the strict requirements of the non-signalling principle remains to be determined. For example, recently, after having found only null results with a new matrix-correlation study [24], one of the original GQT proponents (Walach; see ref. [62]) cautioned that the non-signalling principle might still limit the success of matrix-correlation experiments; specifically, Walach et al. [24] noted that the non-signalling principle which prohibits "… signal transfer in systems that are built on correlations might be operating even in this sophisticated experimental design" (p. 815). Therefore, at least for the psi laboratory studies performed until now, this appears to leave the field of parapsychology with no quantum-based model capable of predicting specific psi effects—whether replicable or not.

In summary, there are—at least—three major constraints limiting the identification of an observed anomalous result as a psi-derived anomaly (Fig 10): (i) The lack of an accepted theoretical framework capable of predicting specific measurement outcomes, (ii) the continuing challenge of implementing appropriate diagnostic calibration procedures, including meaningful negative and positive controls, and (iii) the lack of replicable results in psi research using confirmatory test strategies and preregistration of analysis methods. Finally, because laboratory evidence for anomalous cognition would typically defy the known laws of physics, the next section will briefly discuss how physics addresses the challenges of anomalies research and investigations of unusual phenomena at the frontiers of physics.

## 4.6. Comparing psi research with anomalies research in physics

In contrast to parapsychology or anomalous cognition research, the pursuit of apparently anomalous phenomena in physics, or astrophysics, is typically guided by a strong theory capable of making specific predictions, e.g., see the detection and identification of gravity waves emanating from colliding black-holes [65]. However, if guidance by a theory is unavailable in anomalies research, then the replicability of the anomalous phenomenon by independent investigators becomes the central criterion for establishing scientific credibility. As was mentioned in Section 4.3., an example is the recent report of the detection of an anomalous magnetic moment for muons in high-energy physics, where the collaborative work across many institutions has now come close to confirming the muon anomaly—after decades of joint research [48]. By contrast, again, in the case of anomalous cognition research, neither can any theory predict a specific experimental outcome, nor has a specific anomalous result been shown to be replicable under rigorous test conditions (see Section 1.1.). Therefore, it should come as no surprise that parapsychological claims of positive psi effects are (i) met with strong skepticism by the general scientific community (e.g., [1]) and (ii) referred to, typically, as 'pseudoscience' in academic discussions by the mainstream of science (e.g., [66]). In summary, unless reliably replicable evidence—using confirmatory research designs—could be produced for a particular psi paradigm, it is difficult to envision how new progress could be achieved beyond the exploratory stage in psi research.

As will be discussed next, one radical, alternative option would be to accept the non-replicability of psi laboratory phenomena as an intrinsic feature of all results reported—so far—in the parapsychological literature. This includes both the non-replicability of psi phenomena when testing potential psi abilities with (i) individuals in single-case experimental paradigms as well as (ii) large-scale, collective studies involving many participants as in the present AMP-TPP replication project.

## 4.7. Anomalous non-replicability versus conventional non-replicability

This section introduces the concept of anomalous non-replicability (ANR) as an unconventional hypothesis for explaining non-replicable results in parapsychology. Starting with the results of this replication project, background is provided on how psi investigators have often justified non-replicable results. Note that (i) the here-reported positive finding with confirmatory Study 2 was classified as an ULP (Section 4.4.), and (ii) this replication project found evidence for both replicability and non-replicability using confirmatory test strategies.

For background, some psi-researchers have argued for the view that (i) exploratory (PCH-based) methods enable more frequent observations of psi effects and (ii) the observed non-replicability under confirmatory conditions is itself a psi-derived anomaly; hence, in that view, the observed non-replicability is not the result of conventional factors, such as HARKing [67], confirmation bias [8], or intrinsic measurement bias (e.g., [25]), but is a psi phenomenon itself. For example, the frequent observation that an original finding cannot subsequently be replicated has also become known as the 'Decline Effect'—occasionally implying that the observed non-replicability has an anomalous dimension also (e.g., [68,69]). Other unconventional (psi) ideas to explain the lack of replicability include the already-mentioned concepts of "trickster effect", "psi-missing", or "experimenter psi" (e.g., [13,42–45]). In summary, all these various proposals—in essence—argue that in the area of psi-research, the non-replicability of a finding does not necessarily indicate that this initial finding was a false positive. That position is here referred to as ANR or the ANR-hypothesis. Problematic is, however, the fact that the same two phenomena, i.e., non-replicability and dependence on exploratory (post-hoc) methods, are also the main characteristics of known false-positive results in general scientific research. How to test and prove the ANR-hypothesis? How to distinguish ANR from all types of conventional non-replicability? Is this even possible in a scientifically conclusive manner? To advance new progress on these critical questions, a large research initiative is required for investigating the ANR-hypothesis both theoretically and empirically using advanced methodologies.

As was already mentioned in Section 4.5., the initial attempts to confirm a particular type of ANR with so-called matrix-correlation experiments have so far proven to be unsuccessful (e.g., [18,24]); nevertheless, advanced empirical investigations of the ANR-hypothesis might present the new frontier of anomalous cognition research. In other words, in a first step, researchers would have to prove that the observed non-replicability (i.e., claiming ANR) is empirically distinguishable from all conventional forms of non-replicability—using the established methods of science. This is called a 'first step' because the proposal is based on an Exclusion Principle (for eliminating known conventional explanations for non-replicability) and does not identify—necessarily—the true source of the non-replicability. For explanation, if evidence for a form of ANR could be discovered by convincingly excluding any known conventional accounts of non-replicability, then this still leaves open the possibility of a role for unknown conventional factors and influences—yet to be discovered. Therefore, even in advanced empirical tests of the ANR-hypothesis, researchers must be cautious before making any strong claims about the existence of a psi factor or psi ability, especially in the absence of an accepted theory for explaining any type of psi effect, whether the effect is replicable or not.

For example, regarding tests of the ANR-hypothesis, a large, multi-laboratory replication project might involve an experiment that has repeatedly claimed a substantial (≈5% above chance) psi effect over many decades—the Ganzfeld experiment [e.g., 9,29–32]. Although the Ganzfeld psi effect was not found to be reliably replicable, it was reported that the effect—nevertheless—was replicable in an overall sense, i.e., when considering the available database of studies collectively (e.g., [29,31]). However, given that most of the studies in that database were exploratory in nature, it remains

unknown how many of the results might be false positives, e.g., as a function of simple chance findings or optional stopping, and how many unseen negative findings might there be, e.g., as a function of publication bias and the file-drawer effect. Therefore, using preregistered confirmatory methodologies, a multi-laboratory replication project could be developed—based on the counterfactual AMP-based strategy—for empirically determining the (conventional) false-discovery rate for multiple Ganzfeld psi studies. In short, such a metascientific replication project could study the degree of replicability or—alternatively—type of non-replicability (anomalous versus conventional) of the claimed Ganzfeld psi effect in the four-choice (target-decoy) identification task (e.g., [31]). Novel multi-laboratory replication studies, such as the above proposed project, might be central to a new research initiative for investigating the ANR-hypothesis.

## 4.8. Conclusions and outlook

The methodological advances that were systematically implemented in this metascientific replication project—based on combined AMP [25] and TPP [20] procedures—have contributed to the development of new scientific processes for better distinguishing method-derived from potential psi-derived anomalies (Fig 10). Using this advanced methodology, the present work further disconfirmed the original claim of a precognitive effect in Bem Experiment 1 [35,36]. In the future, the adoption of such advanced research designs, including the use of CFME-based approaches for improving the detection of intrinsic measurement biases, might reduce the risk of mistaking false-positive for true-positive results not only in parapsychological research, but also in other research areas.

The present success of a direct confirmatory replication of an exploratory effect in anomalous cognition research appears to be unique. By comparison, previous preregistered confirmatory replication studies of parapsychological findings have almost exclusively reported null results as was described in Section 1.1. To clarify whether the below-chance effect observed in confirmatory Study 2 (Section 3.2.1.), which was not found to replicable in Study 3 (Section 3.3.), might nevertheless be confirmed with independent replication studies, a multi-laboratory research effort could be conducted regarding this unexpected finding (see also further below regarding the topic of future research).

It can be appreciated why—throughout the history of psi research—some investigators have continued the pursuit of psi phenomena even after decades of failing to gather replicable evidence. Fact is, from time to time, there will appear "statistical" anomalies in certain types of experiments—even if the experiments were well-conducted according to standard scientific norms; however, the crucial question whether a psi-derived anomaly was—in fact—ever detected remains unanswered to this day; to repeat, only if a psi factor could be identified as the true source of even a single anomalous laboratory finding, could parapsychology finally provide an affirmative answer regarding the existence of psi effects under controlled laboratory conditions. Until that time, even a (once-)confirmed anomalous result in a psi-based study can be scientifically described only as an ULP—because of the current inability to reliably distinguish method-derived from psi-derived anomalies (Fig 10).

Regarding future research, the fact that previously-claimed psi effects are increasingly proven to be non-replicable under well-controlled test conditions has provoked the question of whether the observed non-replicability is itself anomalous. This option was discussed in Section 4.7. using the terms ANR and ANR-hypothesis. For testing the ANR-hypothesis, a new research initiative might focus on large-scale, multi-laboratory studies capable of quantitatively assessing the degree of anomalous versus conventional non-replicability. This might well present the final frontier of experimental parapsychology; that is, unless reliable evidence in support of the ANR-hypothesis could be obtained, research on anomalous cognition will likely remain an entirely exploratory program—with no prospect of becoming a recognized science capable of identifying ULPs as psi-derived anomalies using confirmatory protocols.

From a conventional perspective, at a minimum, such an ambitious ANR-research initiative might clarify why—and by way of what scientific processes—some long-investigated paradigms in psi research have yielded statistically-significant anomalous results. In short, if no significant differences between anomalous and conventional rates of non-replicability could be confirmed, then this would confirm—empirically—the inability to scientifically demonstrate the existence of

psi-derived anomalous effects using well-controlled laboratory studies; again, this would also include the class of claimed psi effects that have proven to be non-replicable.

## Acknowledgments

This replication project was commissioned by the Fetzer Franklin Fund of the John E. Fetzer Memorial Trust to be developed and overseen by Phenoscience Laboratories, Berlin, Germany (www.phenoscience.com). We thank Bruce Fetzer for his vision and continuing support of research probing the frontier of science. We thank Sergio Mello e Souza, who is Managing Director of the Scients Institute (https://scients.org), a co-sponsor of the present project, for championing research testing the limits of scientific knowledge. Completion of this metascientific replication project and preparation of this publication was made possible with support by the Paradox Science Institute (www.paradoxscience.org). With thank the reviewers for their thoughtful criticisms and comments which helped greatly improve the presentation of this work.

## Author contributions

**Conceptualization:** Jan Walleczek, Nikolaus von Stillfried, Stefan Schmidt, Marc Wittmann, Zoltan Kekecs.

**Data curation:** Zoltan Kekecs.

**Formal analysis:** Zoltan Kekecs.

**Funding acquisition:** Jan Walleczek, Jorge Moll.

**Investigation:** Jan Walleczek.

**Methodology:** Jan Walleczek, Nikolaus von Stillfried, Stefan Schmidt, Marc Wittmann, Jorge Moll, Zoltan Kekecs.

**Project administration:** Nikolaus von Stillfried, Stefan Schmidt.

**Resources:** Zoltan Kekecs.

**Software:** Zoltan Kekecs.

**Supervision:** Jan Walleczek.

**Validation:** Jan Walleczek, Nikolaus von Stillfried, Stefan Schmidt, Zoltan Kekecs.

**Visualization:** Jan Walleczek.

**Writing – original draft:** Jan Walleczek, Nikolaus von Stillfried, Zoltan Kekecs.

**Writing – review & editing:** Stefan Schmidt, Marc Wittmann, Karolina A. Kirmse, Jorge Moll.

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
