## [Decision Letter · Decision Letter 0]

28 Feb 2025

Dear Dr. Walleczek,

**Editor comment**

We look forward to receiving your revised manuscript.

Kind regards,

Michael B. Steinborn, PhD

Section Editor

PLOS ONE

Journal Requirements:

Reviewers' comments:

Reviewer's Responses to Questions

**Comments to the Author**

1. Is the manuscript technically sound, and do the data support the conclusions?

Reviewer #1: Partly

2. Has the statistical analysis been performed appropriately and rigorously?

Reviewer #1: Yes

3. Have the authors made all data underlying the findings in their manuscript fully available?

Reviewer #1: Yes

4. Is the manuscript presented in an intelligible fashion and written in standard English?

Reviewer #1: Yes

Reviewer #1: The manuscript, “Metascientific replication project with the advanced meta-experimental protocol of the transparent psi project procedures for testing the precognitive effect claimed by Bem,” investigates the replicability of claims about precognitive effects reported by Bem (2011) using advanced methodologies, including the AMP-TPP protocol, across three large-scale studies. While Study 2 replicated an exploratory finding suggesting a slight below-chance effect, neither the original precognitive effect nor the replicated anomaly was confirmed in Study 3. The results highlight methodological advancements for rigor in psychological research and suggest that observed results are better explained by method-derived anomalies rather than psi-theory.

The manuscript features a well-powered design and demonstrates commendable adherence to open-science and transparency practices, significantly enhancing its credibility and reproducibility while offering well-founded criticism of psi-theory. However, several areas warrant improvement, including clarifying key methodological details, improving the comprehensibility of the discussion, and addressing the results more thoroughly with critical consideration of specific psi concepts. Overall, this study is a valuable contribution to the field, and the authors’ commitment to rigorous research practices is evident. I believe that addressing the issues outlined above in a major revision would greatly enhance the manuscript’s quality.

For detailed comments, see the attached document.

**Do you want your identity to be public for this peer review?** For information about this choice, including consent withdrawal, please see our Privacy Policy

Reviewer #1: **Yes: ** Julian Gutzeit

---

## [Author Response · Author response to Decision Letter 1]

26 May 2025

Dear Dr. Steinborn (Section Editor, PLOS ONE),

The attached Response to Reviewer thoroughly addresses all 45 comments as raised by the reviewer. In addition, the substantially revised manuscript is attached also (including in a version showing the revised texts highlighted in red). Also, the figures have been revised and the new figures have been uploaded. Many thanks for your consideration, Jan Walleczek

---

## [Decision Letter · Decision Letter 1]

16 Jul 2025

Dear Dr. Walleczek,

We look forward to receiving your revised manuscript.

Kind regards,

Michael B. Steinborn, PhD

Section Editor

PLOS ONE

Reviewers' comments:

Reviewer's Responses to Questions

**Comments to the Author**

Reviewer #1: All comments have been addressed

Reviewer #2: (No Response)

Reviewer #3: All comments have been addressed

2. Is the manuscript technically sound, and do the data support the conclusions?

Reviewer #1: Yes

Reviewer #2: Yes

Reviewer #3: Partly

3. Has the statistical analysis been performed appropriately and rigorously?

Reviewer #1: Yes

Reviewer #2: Yes

Reviewer #3: Yes

4. Have the authors made all data underlying the findings in their manuscript fully available?

Reviewer #1: Yes

Reviewer #2: No

Reviewer #3: Yes

5. Is the manuscript presented in an intelligible fashion and written in standard English?

Reviewer #1: Yes

Reviewer #2: Yes

Reviewer #3: Yes

Reviewer #1: I want to thank the authors for their thorough response and revision of the manuscript. I think they have successfully improved the clarity and the overall rigor and coherence of their work. I do not require any further changes and recommend accepting the manuscript as it is. Nice work!

Reviewer #2: see my comments below or attached pdf.

General Comments:

As a statistical methodologist and meta-researcher, rather than a domain expert in psi or precognitive effects, my review focuses primarily on the methodological rigor and transparency of the submitted manuscript. I commend the authors for their ambitious effort in designing and conducting three registered replication studies to test the precognitive effect claimed by Bem (2011). The manuscript is well-written, with a clear structure and a commendable commitment to open science practices, including preregistration, data transparency, and the use of advanced methodological protocols such as the Advanced Meta-Experimental Protocol (AMP) and Transparent Psi Project (TPP) procedures. These practices align with the growing emphasis on replicability and reliability in scientific research. Plos One should be proud to consider publishing work that prioritizes such rigor, particularly in a field where replication studies remain underappreciated. The authors’ efforts to address methodological challenges in psi research, such as confirmation bias and intrinsic measurement bias, are noteworthy and contribute to advancing metascience in this domain. However, I have two major concerns regarding the justification for the replication studies and the generalizability of the findings, which warrant major revisions to strengthen the manuscript.

1. The authors provide a critique of the limitations of meta-analyses in parapsychology, noting that the supportive claims for psi phenomena may be compromised by the methodological weaknesses of individual studies included in these analyses (see refs. 9, 32–34). This is a valid concern, given the prevalence of exploratory designs, lack of preregistration, and potential for questionable research practices in the field. However, the rationale for replicating Bem’s Experiment 1 specifically is less robust. The authors state, “The present work investigated the most-widely discussed claim for an experimental psi effect in recent years, namely the precognitive effect claimed with Experiment 1 in the report by Bem. One major reason for choosing the paradigm of Bem Experiment 1 was the fact that a meta-analysis of 90 replication studies claimed an overall significant precognition effect—seemingly confirming the original claim of precognition”. This justification is insufficiently developed, as it does not critically engage with the quality of the 90 replication studies included in Bem et al. (2016; F1000Research, 4, 1188).

Meta-analyses of replication studies, if conducted rigorously, represent high-level evidence, as they aim to synthesize findings from studies explicitly designed to test the same phenomenon, theoretically minimizing selective reporting or publication bias. Questioning the conclusions of such a meta-analysis requires a stronger empirical basis than the general methodological concerns raised in the Introduction. To strengthen their rationale, I recommend that the authors systematically assess the risk of bias in the 90 replication studies included in Bem et al. (2016). This could involve a structured evaluation using established tools (e.g., RoB 2.0) or a custom risk-of-bias framework for replication studies to examine aspects such as preregistration status, sample size adequacy, blinding, and handling of exploratory analyses. The results of this assessment should be reported as an initial section in the Results, providing a clear justification for the need for the current replication studies. Additionally, the authors could explore whether the meta-analytic conclusions depend on study quality by conducting subgroup analyses (e.g., comparing high- vs. low-quality studies) or meta-regression to examine the influence of methodological rigor on effect sizes. Such analyses would provide a more robust foundation for the replication effort and clarify whether the claimed precognitive effect warrants further scrutiny due to methodological flaws in prior studies.

2. The authors have implemented rigorous protocols, including the AMP and TPP, to enhance the validity and transparency of their replication studies. The use of counterfactual meta-experiments (e.g., REG-based trials) and diagnostic calibration routines to detect systematic biases is particularly commendable, as these approaches address potential sources of error that are often overlooked in psychological and parapsychological research. However, the replicability of the authors’ findings—particularly the anomalous below-chance effect observed in Study 2—requires verification by independent laboratories using the same experimental protocol, population characteristics, and analytical methods. The field of metascience has increasingly emphasized the value of multi-laboratory replication projects, such as Registered Replication Reports (RRRs) or Many Labs initiatives, which provide robust evidence of replicability by accounting for laboratory-specific variations and reducing the risk of idiosyncratic findings (e.g., Klein et al., 2014, Perspectives on Psychological Science). While the authors’ three studies are high-powered and methodologically sophisticated, their execution within a single research framework limits the generalizability of the results.

I recommend that the authors acknowledge this limitation explicitly in the Discussion section and tone down their claims about the replicability (or lack thereof) of the precognitive effect. The authors should also discuss the feasibility of a multi-laboratory replication effort to test the robustness of their findings, particularly the unexpected below-chance effect in Study 2, which was not replicated in Study 3. This discussion should address potential sources of heterogeneity (e.g., participant characteristics, online vs. lab settings) that could influence replicability and outline how future studies could address these factors. By moderating their claims and emphasizing the need for independent replication, the authors can strengthen the manuscript’s credibility and align it with best practices in metascience.

Minor comments:

While the statistical analyses are well-described, the rationale for choosing specific priors in the Bayesian proportion tests (e.g., BUJ prior, replication prior) could be clearer. The authors should briefly explain why these priors were selected and how they influence the interpretation of the Bayes factors, particularly for readers unfamiliar with Bayesian methods in parapsychology.

Please make raw data and code publicly available. There are no data and code archived in the repository provided by the authors (https://osf.io/8jteq/)

Sincerely,

Yefeng Yang. Note that I sign all my review comments

Reviewer #3: The article is detailed and of certain reference value, but there are still some minor issues to be addressed, as follows:

1. Description of experimental procedures is not rigorous

"the only substantial differences... participants completed the trials online... minor technical adjustments were made to the software."

The specific content of "minor technical adjustments" is not explained. Please provide an explanation.

2. Discussion section

Sections 4.3-4.7 show a slight imbalance in the comparison between the psi interpretation and the traditional interpretation. Some paragraphs are piled with technical jargon, resulting in insufficient readability.

It is recommended to use a table to compare the core viewpoints, evidential support, and limitations of the two interpretations, so as to enhance visualization.

Simplify the discussion on the quantum mechanics part, focus on the contradiction between the "non-signal transmission principle" and the precognitive effect, and avoid deviating from the theme.

3. In Section 4.8 (Conclusions), add specific suggestions for future research to enhance the forward-looking nature of the study.

4. Please supplement the limitations of this study.

**Do you want your identity to be public for this peer review?** For information about this choice, including consent withdrawal, please see our Privacy Policy

Reviewer #1: **Yes: ** Julian Gutzeit

Reviewer #2: No

Reviewer #3: No

---

## [Author Response · Author response to Decision Letter 2]

1 Sep 2025

I have responded to the Academic Editor in the uploaded Cover Letter. I have responded to the reviewers in the uploaded Response to Reviewers.

---

## [Decision Letter · Decision Letter 2]

9 Oct 2025

Metascientific replication project with the advanced meta-experimental protocol of the transparent psi project procedures for testing the precognitive effect claimed by Bem

PONE-D-24-43107R2

Dear Dr. Walleczek,

We’re pleased to inform you that your manuscript has been judged scientifically suitable for publication and will be formally accepted for publication once it meets all outstanding technical requirements.

Kind regards,

Michael B. Steinborn, PhD

Section Editor

PLOS ONE

Additional Editor Comments (optional):

Reviewer #2:

Reviewer #3:

Reviewers' comments:

Reviewer's Responses to Questions

**Comments to the Author**

Reviewer #2: All comments have been addressed

Reviewer #3: All comments have been addressed

2. Is the manuscript technically sound, and do the data support the conclusions?

Reviewer #2: Partly

Reviewer #3: Yes

3. Has the statistical analysis been performed appropriately and rigorously?

Reviewer #2: Yes

Reviewer #3: Yes

4. Have the authors made all data underlying the findings in their manuscript fully available?

Reviewer #2: Yes

Reviewer #3: Yes

5. Is the manuscript presented in an intelligible fashion and written in standard English?

Reviewer #2: Yes

Reviewer #3: Yes

Reviewer #2: The author team has done a great job in addressing my comments and criticisms, especially in terms of clarifying the motivation of this replication study to properly credit prior work and avoid mis-criticising prior work, and avoid exaggerating findings (generalizability of their findings and conclusions should be interpreted in-context and along with limitations). To clarify, I am not the subject expert, and I am unable to judge the significance of this work. As a statistician, I think this replication study is well-registered, and conducted. I am also a meta-researcher and advocate for open science and encourage more replication studies verifying prior findings. I am delighed to see the publication of this work.

Yefeng Yang (signed)

Reviewer #3: I really appreciate the efforts made by the authors for this. The article has been revised very well. I suggest it be accepted for publication.

**Do you want your identity to be public for this peer review?** For information about this choice, including consent withdrawal, please see our Privacy Policy

Reviewer #2: **Yes: ** Yefeng Yang

Reviewer #3: No

---

## [Editor Report · Acceptance letter]

PONE-D-24-43107R2

PLOS ONE

Dear Dr. Walleczek,

I'm pleased to inform you that your manuscript has been deemed suitable for publication in PLOS ONE. Congratulations! Your manuscript is now being handed over to our production team.

Kind regards,

on behalf of

Dr. Michael B. Steinborn

Section Editor

PLOS ONE